# Explainable GNN-Based Models over Knowledge Graphs

**David Tena Cucala**
Department of Computer Science
University of Oxford, UK

**Bernardo Cuenca Grau**
Department of Computer Science
University of Oxford, UK

**Egor V. Kostylev**
Department of Informatics
University of Oslo, Norway

**Boris Motik**
Department of Computer Science
University of Oxford, UK

## Abstract

Graph Neural Networks (GNNs) are often used to learn transformations of graph data. While effective in practice, such approaches make predictions via numeric manipulations so their output cannot be easily explained symbolically. We propose a new family of GNN-based transformations of graph data that can be trained effectively, but where all predictions can be explained symbolically as logical inferences in Datalog—a well-known rule-based formalism. In particular, we show how to encode an input knowledge graph into a graph with numeric feature vectors, process this graph using a GNN, and decode the result into an output knowledge graph. We use a new class of *monotonic* GNNs (MGNNs) to ensure that this process is equivalent to a round of application of a set of Datalog rules. We also show that, given an arbitrary MGNN, we can automatically extract rules that completely characterise the transformation. We evaluate our approach by applying it to classification tasks in knowledge graph completion.

## 1 Introduction

*Knowledge graphs* (KGs) (Hogan et al., 2021) describe an application domain as a collection of entities and their relationships, and they are frequently used in applications such as Web search, recommendation, fraud detection, and data integration. Many such applications involve realising a transformation on knowledge graphs. For example, in a recommender system, we can represent user-item interactions as a graph (Aggarwal, 2016) and extend it with external KGs to improve accuracy, diversity, and interpretability of recommendations (Zhang et al., 2016; Wang et al., 2018; 2019a); the recommender system can then transform this graph into another graph containing recommended user-item interactions. Moreover, a KG completion system (Rossi et al., 2021; Bordes et al., 2013; Teru et al., 2020) transforms an input graph into another graph containing relationships missing in the input. Usually, such transformations are not given explicitly, but are learned from examples.

*Graph Neural Networks (GNNs)* (Scarselli et al., 2009; Liu & Zhou, 2020) are one of the most popular ML models for graph data. They are usually applied to KGs in three stages (Hamilton et al., 2017). First, the input KG is *encoded* into an *embedding space* as another graph whose vertices are labelled with numeric feature vectors. Second, this graph is processed by a GNN with several layers; for each layer, the GNN updates the feature vectors based on the learned model parameters and the vectors of the previous layer. Finally, the vectors of the output layer are *decoded* to the output KG.

While such approaches are effective in practice, the transformation can be difficult to understand because GNNs compute predictions through numeric calculations. For example, it would be useful to know that a system recommended novel *The Idiot* because the user liked *Crime and Punishment* and both novels were written by Dostoevsky. There is a growing interest in explaining GNN predictions. Numerous methods focus on identifying relevant subgraphs in the embedding space (Ying et al., 2019; Luo et al., 2020; Lin et al., 2021), but such approaches do not explain predictions *symbolically*—that is, by showing how a prediction can be derived from the input KG via logical inferences of a knowledge representation formalism. Datalog (Abiteboul et al.,

1995; Ceri et al., 1989) is a prominent such formalism. It allows one to model an application domain using 'if–then' rules, which can be applied to facts to derive new facts. For example, we can represent knowledge about literature using facts *Author*(*Dostoevsky*, *Crime_and_Punishment*) and *Author*(*Dostoevsky*, *The_Idiot*), and past interactions using fact *Likes*(*Crime_and_Punishment*). Then, applying rule *Author*($x, y_1$) ∧ *Author*($x, y_2$) ∧ *Likes*($y_1$) → *Recommend*($y_2$) to these facts derives fact *Recommend*(*The_Idiot*). Datalog engines (Motik et al., 2014) can generate human-readable proofs explaining why a fact follows from the rules and the data, which can help foster trust in system predictions, ensure norm compliance, and enable verification of fairness standards.

The main contribution of this paper is a new family of GNN-based transformations of KGs that can be trained from examples as usual, but whose predictions can be explained symbolically using Datalog rules. Our transformation consists of the three stages mentioned earlier, but these were carefully crafted to ensure that the transformation can always be described symbolically. To this end, we introduce a novel class of *monotonic GNNs* (MGNNs) that ensure an important property: when values in input feature vectors increase, no value of an output feature vector decreases. To capture this intuition formally, we show that our transformation is *monotonic under homomorphisms*—a well-known property that captures key aspects of Datalog rule application.

In contrast to the related approaches (Teru et al., 2020; Hamaguchi et al., 2017; Schlichtkrull et al., 2018; Wang et al., 2019b), all predictions of our transformation can be explained using Datalog rules. Specifically, for each MGNN, there exists an *equivalent* set of rules such that, on each KG, the rules and the MGNN-based transformation produce the same facts. This opens the door to new ways of using MGNNs: when developing Datalog rules manually is difficult, we can train an MGNN and use it to provide further predictions; however, such predictions can always be explained using rules. For example, we could train an MGNN on a graph containing examples of credit applications and then support decision-making for new applications by applying the model to a different graph. For each MGNN obtained in such a way, there exists a set of rules that *fully* explains each credit recommendation; extracting all such rules can be computationally expensive, but one can typically extract a representative subset of such rules in practice. Thus, we integrate ML and symbolic AI in a single framework, allowing applications to use the best of both worlds.

We demonstrate the effectiveness of our approach by applying it to classification tasks in KG completion. Using well-known benchmarks, we compare our system's performance with the state-of-the-art KG completion systems DRUM (Sadeghian et al., 2019) and AnyBURL (Meilicke et al., 2019). We show that, in addition to providing an exact correspondence between the model and the extracted rules, our technique also offers competitive performance.

Our proofs are given in the appendix, and the source code is available as supplementary material.

## 2 AN MGNN-BASED TRANSFORMATION OF KNOWLEDGE GRAPHS

We now describe our GNN-based transformation of knowledge graphs. In this paper, we see a KG as a *dataset*, which is a finite set of facts of the form $A(a)$ or $R(a_1, a_2)$, where $A$ is a *unary predicate*, $R$ is a *binary predicate*, and $a$, $a_1$, and $a_2$ are *constants*. Constants represent entities from an application domain, unary predicates typically represent entity types, and binary predicates represent entity relationships. For example, the statements that a user likes *Crime and Punishment* and it is a novel written by Fyodor Dostoevsky can be represented as facts *Likes*(*Crime_and_Punishment*) and *Author*(*Dostoevsky*, *Crime_and_Punishment*). KGs in formats such as RDF can be transformed into such a dataset in the obvious way. For $D$ a dataset, $\mathsf{cn}(D)$ is the set of constants occurring in $D$.

Our transformation is realised as an operator $T_{\mathcal{M}}$ that is parameterised by a GNN $\mathcal{M}$ and that maps datasets to datasets. A dataset $D$ is first encoded into a graph $G_D$ whose vertices are labelled with numeric feature vectors; graph $G_D$ is then processed by $\mathcal{M}$; finally, the result is decoded to the output dataset $T_{\mathcal{M}}(D)$. In Section 3, we show that, if $\mathcal{M}$ is a special kind of GNN that we call *monotonic*, then $T_{\mathcal{M}}$ is equivalent to a round of Datalog rule application, and we can automatically extract from $T_{\mathcal{M}}$ an equivalent set of rules. The space of encodings of $D$ consists of *coloured graphs*, which extend standard undirected graphs in two ways. First, each vertex is labelled by a numeric feature vector of a fixed dimension, which allows the graph to be processed by $\mathcal{M}$. Second, each edge in the graph is assigned a colour, which allows us to distinguish different types of connections.

Figure 1: Graphical representation of dataset $D = \{Author(Dostoevsky, Crime\_and\_Punishment),$ $Author(Dostoevsky, The\_Idiot), Likes(Crime\_and\_Punishment)\}$ and its encoding $G_D$

Figure 1 shows the encoding of the dataset $D$ from the introduction. We introduce vertices $v_{Ds}$, $v_{CP}$, and $v_I$ in $G_D$ to represent constants *Dostoevsky*, *The_Idiot*, and *Crime_and_Punishment*. To represent relationships between constants, we introduce vertices $v_{Ds,CP}$, $v_{CP,Ds}$, $v_{Ds,I}$, and $v_{I,Ds}$; we do not introduce vertices $v_{CP,I}$ and $v_{I,CP}$ because the corresponding constants do not occur together in a fact of $D$. We then encode unary and binary predicates using feature vectors labelling the graph's vertices. To this end, we assign to each predicate a fixed position in the feature vectors; for example, we assign position 1 to predicate *Likes* and position 3 to predicate *Author*. Then, we represent fact *Likes(Crime_and_Punishment)* by setting the first element of the feature vector of $v_{CP}$ to one, and we represent fact *Author(Dostoevsky, Crime_and_Punishment)* by setting the third element of the feature vector of $v_{Ds,CP}$ to one. Finally, we connect with coloured edges all pairs of vertices of $G_D$ that refer to related constants. For example, we connect vertices $v_{Ds}$ and $v_{Ds,CP}$ by an edge of colour $c_1$ to indicate that *Dostoevsky* occurs first in the constant pair of $v_{Ds,CP}$. We analogously connect $v_{Ds,CP}$ and $v_{CP}$ by an edge of colour $c_2$. We connect $v_{CP,Ds}$ and $v_{Ds,CP}$ by an edge of colour $c_3$ to indicate that the constant pairs of the two vertices are inverses of each other. Finally, we connect $v_{CP}$ and $v_{Ds}$ by an edge of colour $c_4$ to indicate that the two constants occur jointly in a fact of $D$.

Formally, we fix a set of four *colours* $\mathsf{Col} = \{c_1, c_2, c_3, c_4\}$. We assume that datasets and derived facts will use finitely many predicates, and we fix a list $A_1, \ldots, A_\epsilon, R_{\epsilon+1}, \ldots, R_\delta$ of available unary and binary predicates. We consider vectors and matrices over the reals $\mathbb{R}$ and its nonnegative subset $\mathbb{R}^+$. For $\mathbf{v}$ a vector, $(\mathbf{v})_i$ is the $i$-th element of $\mathbf{v}$. We apply scalar functions to vectors and matrices element-wise; for example, for $\mathbf{v}_1, \ldots, \mathbf{v}_n$ vectors of equal dimension, $\max\{\mathbf{v}_1, \ldots, \mathbf{v}_n\}$ is the vector whose $i$-th element is $\max\{(\mathbf{v}_1)_i, \ldots, (\mathbf{v}_n)_i\}$. Finally, a function $\sigma : \mathbb{R} \to \mathbb{R}$ is *monotonically increasing* if $x < y$ implies $\sigma(x) \le \sigma(y)$, and function $\sigma$ is *Boolean* if it ranges over $\{0, 1\}$. The following definition describes the kind of graph that our encoding produces.

**Definition 1.** *A* $(\mathsf{Col}, \delta)$-*graph is a tuple* $\langle V, \{E_c\}_{c \in \mathsf{Col}}, \lambda \rangle$ *where $V$ is a finite set of vertices; for each $c \in \mathsf{Col}$, $E_c$ is a set of $c$-coloured undirected edges (possibly including self-loops) over $V$; and labelling $\lambda$ assigns to each $v \in V$ a vector $\lambda(v)$ of dimension $\delta$. If $\lambda$ is implicitly understood, $\lambda(v)$ is written as $\mathbf{v}$. For $c \in \mathsf{Col}$ and $v \in V$, $N_c(v)$ is the set of vertices connected to $v$ by an edge in $E_c$.*

We are now ready to define our encoding of an arbitrary dataset $D$ into a $(\mathsf{Col}, \delta)$-graph $G_D$.

**Definition 2.** *The* encoding *of a dataset $D$ is the* $(\mathsf{Col}, \delta)$-*graph $G_D$ that contains a vertex $v_a$ for each constant $a \in \mathsf{cn}(D)$, and vertices $v_{a,b}$ and $v_{b,a}$ for all (not necessarily distinct) constants $a$ and $b$ that occur together in a binary fact in $D$. For each vertex $v_{a,b}$, $G_D$ contains an edge of colour $c_1$ connecting $v_{a,b}$ and $v_a$, an edge of colour $c_2$ connecting $v_{a,b}$ and $v_b$, an edge of colour $c_3$ connecting $v_{a,b}$ and $v_{b,a}$, and an edge of colour $c_4$ connecting $v_a$ and $v_b$. For each $v_a$, labelling $(\mathbf{v}_a)_i$ is 1 if $A_i(a) \in D$ and 0 otherwise. For each $v_{a,b}$, labelling $(\mathbf{v}_{a,b})_i$ is 1 if $R_i(a, b) \in D$ and 0 otherwise.*

Graph $G_D$ will be processed by a GNN. As we explain in detail in Section 3, if we are to ensure that our transformation 'mimics' Datalog rule application, we cannot use an arbitrary GNN. Instead, we introduce *Monotonic Graph Neural Networks (MGNNs)*, which restrict GNNs in a way that allows us to attain our objective. Definition 3 also specifies how to apply an MGNN to a coloured graph.

**Definition 3.** *A* $(\mathsf{Col}, \delta)$-monotonic graph neural network *(MGNN)* $\mathcal{M}$ *with $L \ge 1$ layers is a tuple* $\langle \{\mathbf{A}^\ell\}_{1 \le \ell \le L}, \{\mathbf{B}_c^\ell\}_{c \in \mathsf{Col}, 1 \le \ell \le L}, \{\mathbf{b}^\ell\}_{1 \le \ell \le L}, \sigma, \mathsf{cls} \rangle$. *For each $\ell \in \{1, \ldots, L\}$ and each $c \in \mathsf{Col}$, $\mathbf{A}^\ell$ and $\mathbf{B}_c^\ell$ are matrices over $\mathbb{R}^+$ of dimension $m^\ell \times n^\ell$, and $\mathbf{b}^\ell$ is a vector over $\mathbb{R}$ of dimension $n^\ell$ such that $n^1 = m^L = \delta$, and $m^{\ell-1} = n^\ell$ for each $\ell \in \{2, \ldots, L\}$. Moreover, $\sigma : \mathbb{R} \to \mathbb{R}^+$ is a monotonically increasing activation function, and $\mathsf{cls} : \mathbb{R}^+ \to \{0, 1\}$ is a monotonically increasing classification function. The dimension of $\mathcal{M}$ is $\max\{n^1, \ldots, n^L\}$.*

*Applying $\mathcal{M}$ to a $(\mathsf{Col}, \delta)$-graph $G = \langle V, \{E_c\}_{c \in \mathsf{Col}}, \lambda \rangle$ induces the sequence $\lambda^0, \ldots, \lambda^L$ of vertex labellings where $\lambda^0 = \lambda$ and, for each $\ell \in \{1, \ldots, L\}$ and $v \in V$, the value of $\lambda^\ell(v)$ is given by*

$$\mathbf{v}^\ell = \sigma\big(\mathbf{A}^\ell \mathbf{v}^{\ell-1} + \sum_{c \in \mathsf{Col}} \mathbf{B}_c^\ell \max\{\mathbf{w}^{\ell-1} \mid w \in N_c(v)\} + \mathbf{b}^\ell\big). \tag{1}$$

*The result $\mathcal{M}(G)$ of applying $\mathcal{M}$ to $G$ is the $(\mathsf{Col}, \delta)$-graph with the same vertices and edges as $G$, but where each vertex $v \in V$ is labelled by $\mathsf{cls}(\lambda^L(v))$.*

Intuitively, an MGNN consists of $L$ layers, where each layer $\ell \in \{1, \ldots, L\}$ is defined by a matrix $\mathbf{A}^\ell$, a matrix $\mathbf{B}_c^\ell$ for each colour $c$, and a bias vector $\mathbf{b}^\ell$; these constitute the learnable parameters of the model. In addition, an MGNN also uses functions $\sigma$ and $\mathsf{cls}$ which are fixed and explicitly given. An MGNN transforms an arbitrary $(\mathsf{Col}, \delta)$-graph $G$ into another $(\mathsf{Col}, \delta)$-graph $\mathcal{M}(G)$ as follows. For each layer $\ell \in \{0, \ldots, L\}$, each vertex $v$ in $G$ is associated with a labelling $\mathbf{v}^\ell$. Labelling $\mathbf{v}^0$ is the labelling $\mathbf{v}$ of $v$ in $G$, and, for each layer $\ell$, vertex $v$ is labelled by a feature vector $\mathbf{v}^\ell$ computed as specified in equation (1). The feature vector of $v$ in the resulting graph is computed by applying the classification function $\mathsf{cls}$ to the final labelling $\mathbf{v}^L$. Thus, MGNNs specialise GNNs by (i) taking the maximum of the features of a vertex' neighbours (instead of the more commonly used sum or average), (ii) using nonnegative weights in matrices $\mathbf{A}^\ell$ and $\mathbf{B}_c^\ell$ (but the elements of bias vectors $\mathbf{b}^\ell$ can be negative), and (iii) requiring the activation and classification functions to be monotonically increasing. We discuss the reasons for these restrictions in Section 3.

The result $\mathcal{M}(G_D)$ of applying an MGNN to the encoding $G_D$ of a dataset $D$ can be decoded into the output dataset of the transformation by essentially inverting the encoder (cf. Figure 1). Thus, for each MGNN $\mathcal{M}$, we obtain an operator $T_{\mathcal{M}}$ that is specified as follows.

**Definition 4.** *For $\mathcal{M}$ a $(\mathsf{Col}, \delta)$-MGNN and $D$ a dataset, $T_{\mathcal{M}}(D)$ is the dataset that contains the fact $A_i(a)$ for each $i \in \{1, \ldots, \epsilon\}$ and each vertex $v_a$ such that $(\mathbf{v}_a)_i = 1$ in $\mathcal{M}(G_D)$, and the fact $R_i(a, b)$ for each $i \in \{\epsilon + 1, \ldots, \delta\}$ and each vertex $v_{a,b}$ such that $(\mathbf{v}_{a,b})_i = 1$ in $\mathcal{M}(G_D)$.*

Note that the dimensions of the matrices of $\mathcal{M}$ are independent from the number of constants in a dataset; in fact, $\mathcal{M}$ can be trained on one dataset and then applied to another dataset that contains the same predicates but completely different constants. Our transformation is thus ideally suited for *inductive settings* such as KG completion and recommendation (Teru et al., 2020; Hamaguchi et al., 2017; van den Berg et al., 2017), where training and testing can involve distinct entities.

## 3 EXTRACTING AN EQUIVALENT SET OF DATALOG RULES

We now show that, for each MGNN $\mathcal{M}$, we can automatically generate a set of Datalog rules $\mathcal{P}_{\mathcal{M}}$ such that $T_{\mathcal{M}}$ and $\mathcal{P}_{\mathcal{M}}$ derive the same facts on any dataset $D$. To this end, we first recapitulate the basic definitions of Datalog and identify a property that characterises Datalog reasoning.

A (Datalog) *atom* is of the form $A(t_1)$ or $R(t_1, t_2)$, where $A$ and $R$ are unary and binary predicates, respectively, and each $t_i$ is a constant or *variable*. A (Datalog) *rule* is an implication of the form (2), where $H$ is the atom called the rule's *head*, all $B_i$ are atoms that together form the rule's *body*, and each variable in $H$ occurs in some $B_i$.

$$H \leftarrow B_1 \wedge \cdots \wedge B_m \tag{2}$$

A (Datalog) *program* is a finite set of rules. A *substitution* $\nu$ is a mapping of finitely many variables to constants; for $\alpha$ an atom, $\alpha\nu$ is the result of replacing in $\alpha$ each variable $x$ on which $\nu$ is defined with $\nu(x)$. Each rule $r$ of form (2) defines the *immediate consequence* operator $T_r$ on datasets: for $D$ a dataset, set $T_r(D)$ contains fact $H\nu$ for each substitution $\nu$ satisfying $\{B_1\nu, \ldots, B_m\nu\} \subseteq D$. For $\mathcal{P}$ a program, operator $T_{\mathcal{P}}$ on datasets is defined by $T_{\mathcal{P}}(D) = \bigcup_{r \in \mathcal{P}} T_r(D)$.

Our objective is to show that, for each MGNN $\mathcal{M}$, operator $T_{\mathcal{M}}$ 'behaves analogously' to Datalog rule application. Definition 5 specifies formally what this means in our setting.

**Definition 5.** *A homomorphism from a dataset $D$ to a dataset $D'$ is a mapping $h$ of constants to constants that is defined at least on $\mathsf{cn}(D)$ and that satisfies $h(D) \subseteq D'$, where $h(D)$ is the dataset obtained by replacing each constant $a$ in $D$ with $h(a)$ and removing all duplicate facts. An operator $T$ from datasets to datasets is monotonic under homomorphisms if, for all datasets $D$ and $D'$, each homomorphism from $D$ to $D'$ is also a homomorphism from $T(D)$ to $T(D')$.*

For any constant-free program $\mathcal{P}$, operator $T_{\mathcal{P}}$ is monotonic under homomorphisms. For example, let $D = \{A(a), R(a, b)\}$ and $\mathcal{P} = \{S(x, y) \leftarrow A(x) \wedge R(x, y)\}$, so $T_{\mathcal{P}}(D) = \{S(a, b)\}$. First, operator $T_{\mathcal{P}}$ is *monotonic*: adding facts to $D$ cannot invalidate a consequence of $T_{\mathcal{P}}$ because the body of $r$ still matches so $r$ derives $S(a, b)$. Second, operator $T_{\mathcal{P}}$ is resilient to replacement of constants. For example, let $D' = \{A(c), R(c, c)\}$ be obtained from $D$ by replacing $a$ and $b$ with $c$. Rule $r$ 'cannot tell' that the constants have been replaced; thus, it derives $T_{\mathcal{P}}(D') = \{S(c, c)\}$, which is obtained from $T_{\mathcal{P}}(D)$ by the same replacement. Monotonicity under homomorphisms combines both properties. We next show that rules can be extracted from any operator satisfying this property.

**Definition 6.** *An operator $T$ on datasets* captures *a rule or program $\alpha$ if $T_{\alpha}(D) \subseteq T(D)$ for each dataset $D$.*

**Proposition 7.** *Let $T$ be an operator on datasets, let $r$ be a constant-free rule of form* (2)*, let $\mu$ be a substitution that maps each variable of $r$ to a distinct constant, and let $D_r = \{B_1\mu, \ldots, B_m\mu\}$. If $T$ is monotonic under homomorphisms and $H\mu \in T(D_r)$, then $T$ captures $r$.*

Intuitively, if some operator $T$ captures a rule $r$, then $T_r$ provides us with a *lower bound* for the consequences of $T$ on an arbitrary dataset—that is, $T$ necessarily derives at least as many facts as $r$ (but $T$ can derive more facts). Moreover, if $T$ is monotonic under homomorphisms, Proposition 7 provides us with an effective way of checking whether $T$ captures $r$: we replace each variable in $r$ with a distinct constant, we collect the resulting body atoms in a dataset $D_r$, we apply $T$ to $D_r$, and we check whether the result contains the replaced head atom of $r$. Clearly, $T$ captures a constant-free program $\mathcal{P}$ if $T$ captures each rule $r \in \mathcal{P}$, so we can just apply this check to each $r \in \mathcal{P}$ to verify whether $T$ captures $\mathcal{P}$. This provides us with the necessary machinery for rule extraction.

We next show that operator $T_{\mathcal{M}}$ is monotonic under homomorphisms. This shows that $T_{\mathcal{M}}$ indeed 'behaves like' Datalog reasoning, and it will allow us to use Proposition 7 for rule extraction.

**Theorem 8.** *For each* $(\mathsf{Col}, \delta)$*-MGNN $\mathcal{M}$, mapping $T_{\mathcal{M}}$ is monotonic under homomorphisms.*

Theorem 8 is a consequence of the restrictions on MGNNs mentioned after Definition 3. For monotonicity, note that adding facts to a dataset $D$ has the effect of possibly adding new vertices and edges and then flipping some elements of feature vectors of $G_D$ from 0 to 1; moreover, the elements of $\mathbf{A}^{\ell}$ and $\mathbf{B}_c^{\ell}$ are nonnegative so the arguments of $\sigma$ in equation (1) can only increase; finally, functions $\sigma$ and $\mathsf{cls}$ are monotonically increasing so the elements of the feature vectors of $\mathcal{M}(G_D)$ can only flip from 0 to 1, which is equivalent to deriving more facts in $T_{\mathcal{M}}(D)$. For the resilience of constant merging, replacing, say, constant $b$ with constant $a$ in $D$ is tantamount to merging vertices $v_b$ into $v_a$ in $G_D$; this will cause vertex $v_a$ to have fewer neighbours, but since we aggregate neighbours' feature vectors by taking the maximum, this cannot change the result of equation (1). Note that, if we aggregated neighbours using a sum (as is common in the literature), then elements of the feature vectors in $\mathcal{M}(G_D)$ could decrease due to merging, which would invalidate Theorem 8.

Theorem 8 tells us that we can check whether $T_{\mathcal{M}}$ captures a rule $r$ using Proposition 7. This is a key step of our rule extraction algorithm. In particular, we next show that it is sufficient to consider only rules of a certain structure, and that the number of such rules is determined by the number of predicates $\delta$, the dimension of $\mathcal{M}$, and the number of layers of $\mathcal{M}$. Thus, to obtain a program $\mathcal{P}_{\mathcal{M}}$ deriving the same consequences as $T_{\mathcal{M}}$, we can, in principle, enumerate all such rules, apply Proposition 7 to each rule, and keep the rules that pass this check. Such a naïve procedure can be optimised in practice. For example, we can reduce the number of considered rules by observing that rules can logically imply other rules; for example, if we determine that a rule $B(x) \leftarrow A(x)$ is captured, then we do not need to consider any rule of the form $B(x) \leftarrow A(x) \wedge C$ where $C$ is an arbitrary conjunction: each such rule is logically implied by $B(x) \leftarrow A(x)$ and is thus redundant.

Definition 9 describes the syntactic structure of the rules that can fully characterise $T_{\mathcal{M}}$. To this end, we use the well-known notion of a *primal graph* of a conjunction $C$ of atoms, which is the undirected graph (possibly with self-loops) whose vertices are the variables of $C$, and where an edge connects each pair of (not necessarily distinct) variables $x$ and $y$ such that $R(x, y)$ is in $C$ for some $R$. Note that the primal graph of an empty conjunction is also empty. A primal graph is a (possibly empty) *tree* if it is connected and acyclic (and, in particular, without self-loops).

**Definition 9.** *Let $c$ and $d$ be nonnegative integers. A constant-free conjunction $C$ is $(c, d)$-tree-like for a variable $x$ if the primal graph of $C$ is a tree that is either empty or can be rooted at $x$ (i.e., $x$ occurs in $C$) and, for each vertex $v$ in the tree, the distance $i$ of $v$ from $x$ is at most $d$ and*

*moreover $v$ has at most $c(d - i)$ children. A $(c, d)$-tree-like rule is a rule of form $A(x) \leftarrow C_x$ or $R(x, y) \leftarrow C_x \wedge B_{x,y} \wedge C_y$, where $C_x$ and $C_y$ are $(c, d)$-tree-like conjunctions for $x$ and $y$, respectively, with no variables in common, and $B_{x,y}$ is a nonempty constant-free conjunction of binary atoms of the form $S(x, y)$ or $S(y, x)$.*[1]

*For $\mathcal{M}$ a $(\mathsf{Col}, \delta)$-MGNN with $L$ layers and dimension $k$ (see Definition 3), $\mathcal{P}_\mathcal{M}$ is the program that contains, up to variable renaming, each $(3k, L)$-tree-like rule that is captured by $T_\mathcal{M}$.*

The primal graph of conjunction $A(x) \wedge R(x, y) \wedge S(x, z)$ is a tree with root $x$ and children $y$ and $z$; thus, it is $(2,1)$-tree like for $x$, but not $(1,1)$-tree-like for $x$ (since $x$ is at distance 0 from itself it can have at most one child). In contrast, the primal graph of $A(x) \wedge R(x, y) \wedge S(x, z) \wedge T(y, z)$ is a triangle with nodes $x$, $y$, and $z$, so it is not $(c, d)$-tree-like for $x$ for any $c$ and $d$.

**Theorem 10.** *For each $(\mathsf{Col}, \delta)$-MGNN $\mathcal{M}$ and each dataset $D$, we have $T_\mathcal{M}(D) = T_{\mathcal{P}_\mathcal{M}}(D)$.*

To prove Theorem 10, we present a construction that, for each $(\mathsf{Col}, \delta)$-MGNN $\mathcal{M}$, dataset $D$, and fact $\alpha \in T_\mathcal{M}(D)$, constructs a rule $r \in \mathcal{P}_\mathcal{M}$ such that $\alpha \in T_r(D)$—that is, rule $r$ explains symbolically how $\mathcal{M}$ derives $\alpha$ on $D$. Whether such an explanation is minimal is an open question.

Theorem 10 also describes the expressiveness (i.e., the kind of possible predictions) of $T_\mathcal{M}$. For example, no MGNN can capture rule $r = R(x, z) \leftarrow R(x, y) \wedge R(y, z)$. To see that this is the case, let $D = \{R(a, b), R(b, c)\}$. For an arbitrary MGNN $\mathcal{M}$, Theorem 10 ensures that all predictions of $T_\mathcal{M}$ involving binary facts are made by rules of the form $R(x, y) \leftarrow C_x \wedge B_{x,y} \wedge C_y$; however, no such rule can derive $R(a, c)$ on $D$, so we necessarily have $R(a, c) \notin T_\mathcal{M}(D)$ as well.

## 4 EVALUATION

We evaluated our approach on KG completion—the problem of extending a KG seen as a dataset $D$ to its complete version $D'$ by adding missing facts. When seen as a classification problem, the aim is to learn a Boolean function $f(\cdot, \cdot)$ that takes a dataset $D$ and a fact $\alpha$ over a fixed set of unary and binary predicates such that $f(D, \alpha)$ returns *true* if and only if $\alpha \in D'$. We consider the *inductive* variant of this problem, where the testing dataset may contain constants not occurring in the training or validation datasets (but all predicates in the testing dataset are known before training). Inductive KG completion is more general and challenging than the *transductive* variant (Bordes et al., 2013), where all constants in the testing data also occur in the training data.

**Baselines.** We compared our approach with DRUM (Sadeghian et al., 2019) and AnyBURL (Meilicke et al., 2019), two state-of-the-art approaches for knowledge graph completion.

DRUM learns an end-to-end differentiable model that, given a dataset $D$ and a fact $\alpha$, returns a score representing the likelihood of $\alpha$ belonging to the completion of $D$; predictions are then computed by applying a threshold to the score. This is achieved by learning *chain* rules of the form $H(x_1, x_n) \leftarrow B_1(x_0, x_1) \wedge \cdots \wedge B_n(x_{n-1}, x_n)$, each associated with a confidence score.

AnyBURL uses path sampling techniques to produce a more general variant of chain rules in which variables can be replaced by constants. Each rule is associated with a score. To compute a prediction value for a dataset $D$ and fact $\alpha$, AnyBURL applies all rules to $D$ once, combines the confidence scores of all rules that derive $\alpha$, and applies a threshold to the result.

**Benchmarks and performance metrics.** We used the 12 KG completion benchmarks by Teru et al. (2020), which are based on FB15K-237 (Bordes et al., 2013), NELL-995 (Xiong et al., 2017), and WN18RR (Dettmers et al., 2018). Each benchmark provides disjoint datasets $\mathcal{T}$, $\mathcal{V}$, and $\mathcal{S}$ for training, validation, and testing, respectively; dataset statistics are shown in Table 1.

We evaluated our systems as follows. Each benchmark provides a method for randomly splitting its testing dataset $\mathcal{S}$ into the incomplete dataset $\mathcal{S}_I$ and the set $\mathcal{S}_M$ of missing facts that should be added to $\mathcal{S}_I$. We took $(\mathcal{S}_I, \alpha)$ for each fact $\alpha \in \mathcal{S}_M$ as a positive example for testing. Moreover, we randomly sampled $|\mathcal{S}_M|$ facts $\beta_1, \ldots, \beta_n$ that use the predicates of $\mathcal{S}$ but are not contained in $\mathcal{S}_M$, and we took each $(\mathcal{S}_I, \beta_i)$ as a negative example for testing. Sampling was necessary because the set of possible facts not contained in $\mathcal{S}$ is very large; moreover, random sampling of negative examples

---

[1]Note that $C_x$ cannot be empty in a rule of form $A(x) \leftarrow C_x$ since $x$ is required to occur in the body.

| | FB15K-237 | | | | NELL-995 | | | | WN18RR | | | |
|---|---|---|---|---|---|---|---|---|---|---|---|---|
| | v1 | v2 | v3 | v4 | v1 | v2 | v3 | v4 | v1 | v2 | v3 | v4 |
| Train. | 4,245 | 9,739 | 17,986 | 27,203 | 4,687 | 8,219 | 16,393 | 7,546 | 5,410 | 15,262 | 25,901 | 7,940 |
| Valid. | 489 | 1,166 | 2,194 | 3,352 | 414 | 922 | 1,851 | 876 | 630 | 1,838 | 3,097 | 934 |
| Test. | 2,198 | 4,623 | 8,271 | 13,138 | 933 | 5,062 | 8,857 | 7,804 | 1,806 | 4,452 | 6,932 | 13,763 |

Table 1: Number of facts for each benchmark and phase (Training, Validation, and Testing)

| | | Precision | | | Recall | | | Accuracy | | | F1 Score | | | AUC | | | Training (s) | |
|---|---|---|---|---|---|---|---|---|---|---|---|---|---|---|---|---|---|---|
| | | A | D | M | A | D | M | A | D | M | A | D | M | A | D | M | D | M |
| FB15K-237 | v1 | **100.0** | 82.2 | 98.5 | 25.6 | **40.5** | 32.7 | 62.9 | 65.9 | **66.1** | 41.1 | **54.2** | 49.1 | 62.0 | **68.6** | 65.7 | 15,540 | 9,192 |
| | v2 | **100.0** | 85.3 | 99.5 | 47.9 | **48.5** | 41.2 | **74.0** | 70.0 | 70.5 | **64.5** | 61.9 | 58.3 | 73.8 | **76.0** | 70.4 | 19,080 | 23,523 |
| | v3 | **100.0** | 85.6 | 99.0 | 43.6 | **45.2** | 34.6 | **71.2** | 68.8 | 67.1 | **60.7** | 59.2 | 51.2 | 71.7 | **73.3** | 67.4 | 37,776 | 45,104 |
| | v4 | **100.0** | 88.8 | 99.4 | **46.0** | 41.6 | 35.1 | **73.0** | 68.2 | 67.5 | **63.0** | 56.7 | 52.0 | 73.0 | **73.7** | 67.3 | 87,000 | 42,540 |
| NELL-995 | v1 | 97.5 | 94.7 | 96.4 | 77.0 | 18.0 | **80.0** | 87.5 | 58.5 | **88.5** | 86.0 | 30.3 | **87.4** | 85.7 | 52.6 | **89.6** | 402 | 662 |
| | v2 | **100.0** | 80.7 | **100.0** | **53.1** | 41.2 | 48.3 | **76.6** | 65.7 | 74.2 | **69.4** | 54.5 | 65.2 | **76.3** | 74.6 | 74.2 | 6,354 | 3,575 |
| | v3 | **100.0** | 85.8 | **100.0** | 47.3 | 45.6 | **55.3** | 73.7 | 69.0 | **77.6** | 64.2 | 59.6 | **71.2** | 69.3 | 75.3 | **77.6** | 36,090 | 13,456 |
| | v4 | **100.0** | 80.3 | 99.5 | 44.3 | 26.8 | **57.2** | 72.2 | 60.1 | **78.5** | 61.4 | 40.2 | **72.6** | 71.6 | 67.1 | **78.5** | 3,990 | 16,839 |
| WN18RR | v1 | 99.1 | 97.9 | **100.0** | 58.5 | **73.4** | 62.8 | 79.0 | **85.9** | 81.4 | 73.6 | **83.4** | 77.1 | 78.8 | **92.5** | 80.1 | 312 | 292 |
| | v2 | **100.0** | 96.3 | **100.0** | **75.7** | 69.8 | 60.8 | **87.9** | 83.6 | 80.4 | **86.2** | 80.9 | 75.6 | 59.8 | **87.4** | 80.3 | 2,856 | 856 |
| | v3 | 99.7 | 91.3 | **100.0** | 48.4 | **59.7** | 28.3 | 74.1 | **77.0** | 64.1 | 65.2 | **72.2** | 44.1 | 60.6 | **85.2** | 64.0 | 9,984 | 2,423 |
| | v4 | 99.9 | 98.3 | **100.0** | **71.6** | 65.6 | 58.0 | **85.8** | 82.2 | 79.0 | **83.4** | 78.7 | 73.4 | 59.0 | **93.9** | 79.0 | 1,638 | 409 |

Table 2: Results for AnyBURL (A), DRUM (D), and our MGNN-based system (M), in percentages

for testing does not favour any particular way of constructing negative examples for training. The number of positive and negative examples was the same to ensure a balance between precision and recall. For each system, we classified the positive and negative testing examples and computed the precision, recall, accuracy, and F1 score in the usual way, as well as the area under the precision-recall curve (AUC) by considering different classification thresholds between 0.01 and 0.99.

**Training.** Our system is trained as a denoising autoencoder (Vincent et al., 2010). The training dataset $\mathcal{T}$ was split with a 9:1 ratio into an incomplete dataset $\mathcal{T}_I$ and a set $\mathcal{T}_M$ of missing facts that should be added to $\mathcal{T}_I$. We used $(\mathcal{T}_I, \alpha)$ for each fact $\alpha \in \mathcal{T}_M$ as a positive example for training, and we obtained negative examples $(\mathcal{T}_I, \beta)$ by sampling $\beta$ analogously to how this was done for testing. We trained MGNNs with two layers and the ReLU activation function; we used cross-entropy loss with a logistic sigmoid on $\lambda^L(v)$ as the output probability; finally, we used the Adam optimisation algorithm with the standard learning rate $(0.01)$ and weight decay $(5 \times 10^{-4})$, and a maximum of $50,000$ epochs. We implemented the procedure in Pytorch Geometric v1.5.0, and we ran it on a laptop running macOS 10.15.7 with 8 GB of RAM and an Intel Core i5 2.30 GHz CPU.

We trained the baseline systems using their (publicly available) code bases and their default configurations. For each system and benchmark, we computed the accuracy on the validation dataset for a range of thresholds; in our tests, we used the threshold that maximised accuracy.

**Evaluation results.** Table 2 shows the results of our evaluation. As one can see, our system was very competitive, and it outperformed both baselines on most benchmarks based on NELL-995. Table 2 also shows the training times for DRUM and our system, which are are comparable; we cannot show any times for AnyBURL since this system does not train a model.

Similarly to AnyBURL, our system attained extremely high precision scores; however, recall values were significantly lower, especially for the benchmarks based on FB15K. This is because the testing dataset $\mathcal{S}_M$ was obtained from $\mathcal{S}$ by random splitting so it contains many facts of the form $R(a, b)$ where constants $a$ and $b$ do not occur together in any fact in $\mathcal{S}_I$. By Theorem 10 from Section 3, our transformation can never produce such facts in the output. To confirm our hypothesis that this is the main reason for low recall scores, we additionally split the testing dataset $\mathcal{S}$ so that a fact $R(a, b)$ is placed into $\mathcal{S}_M$ only if $\mathcal{S}_I$ contains a fact of the form $S(a, b)$ or $S(b, a)$ for some predicate $S$. The results for our system obtained in this way are shown in Table 3. As one can see, precision scores remain high, whereas recall values increase significantly; thus, accuracy, F1 score, and AUC values improve accordingly. Our system significantly outperformed all baselines in this setting; however,

| Benchmark | | Precision | Recall | Accuracy | F1 Score | AUC |
|---|---|---|---|---|---|---|
| FB15K-237 | v1 | 95.5 | 74.0 | 85.3 | 83.4 | 89.5 |
| | v2 | 90.3 | 70.0 | 79.9 | 76.9 | 85.8 |
| | v3 | 88.8 | 72.6 | 81.7 | 79.9 | 86.9 |
| | v4 | 90.5 | 79.7 | 85.6 | 84.7 | 91.7 |
| NELL-995 | v1 | 92.3 | 98.8 | 95.2 | 95.5 | 93.3 |
| | v2 | 83.5 | 57.3 | 73.0 | 68.0 | 79.4 |
| | v3 | 82.3 | 71.1 | 77.9 | 76.3 | 89.0 |
| | v4 | 78.5 | 62.4 | 72.7 | 69.5 | 75.8 |
| WN18RR | v1 | 82.1 | 100.0 | 89.1 | 90.2 | 95.1 |
| | v2 | 91.7 | 98.0 | 94.6 | 94.7 | 98.7 |
| | v3 | 88.2 | 98.9 | 92.9 | 93.3 | 89.0 |
| | v4 | 83.8 | 99.0 | 89.9 | 90.8 | 97.7 |

Table 3: Evaluation of the MGNN transformation with alternative splitting of the testing dataset

| Benchmark | | # Rules with... | |
|---|---|---|---|
| | | 1 body atom | 2 body atoms |
| FB15K-237 | v1 | 2,102 | 41,341 |
| | v2 | 3,370 | 27,900 |
| | v3 | 3,199 | 42,590 |
| | v4 | 8,464 | 82,003 |
| NELL-995 | v1 | 149 | 215 |
| | v2 | 1,784 | 3,087 |
| | v3 | 2,308 | 5,085 |
| | v4 | 1,308 | 4,348 |
| WN18RR | v1 | 19 | 3 |
| | v2 | 22 | 35 |
| | v3 | 23 | 19 |
| | v4 | 20 | 54 |

Table 4: Rules of $\mathcal{P}_{\mathcal{M}}$ extracted

| Benchmark | | Total AnyBURL | Tree-like AnyBURL | Captured by $T_{\mathcal{M}}$ | Captured on $\mathcal{S}_I$ |
|---|---|---|---|---|---|
| FB15K-237 | v1 | 2812 | 252 | 72 | 1371 |
| | v2 | 2823 | 404 | 102 | 1048 |
| | v3 | 4805 | 490 | 118 | 1749 |
| | v4 | 3574 | 584 | 134 | 1046 |
| NELL-995 | v1 | 891 | 51 | 49 | 707 |
| | v2 | 1123 | 181 | 112 | 650 |
| | v3 | 1104 | 277 | 181 | 603 |
| | v4 | 841 | 187 | 108 | 606 |
| WN18RR | v1 | 397 | 15 | 11 | 30 |
| | v2 | 522 | 21 | 14 | 35 |
| | v3 | 424 | 21 | 21 | 48 |
| | v4 | 511 | 16 | 12 | 23 |

Table 5: Comparison of rule extraction with AnyBURL

this way of splitting the data gives our system an unfair advantage in a comparison, so we refrain from including in Table 3 the results of the baselines on these alternative testing datasets.

**Rule Extraction.** For each benchmark, we computed using Algorithm 1 in Appendix B the subset $\mathcal{P} \subseteq \mathcal{P}_{\mathcal{M}}$ of all nonredundant rules with at most two body atoms. Table 4 shows the size of each such $\mathcal{P}$. Operator $T_{\mathcal{P}}$ thus approximates $T_{\mathcal{M}}$—that is, $T_{\mathcal{P}}(D) \subseteq T_{\mathcal{M}}(D)$ holds for each dataset $D$. To determine the quality of the approximation, we computed the precision and recall of $T_{\mathcal{P}}$ on the test dataset of each benchmark; Table 6 in Appendix C shows the complete results. One can see that precision is largely the same as for $T_{\mathcal{M}}$, while recall decreases slightly (1.6% on average). Thus, 'short' rules seem to account for almost all predictions on the benchmarks.

We analysed $\mathcal{P}$ manually and noticed that our transformation was able to learn numerous sound rules such as symmetry (e.g., $sibling(x, y) \leftarrow sibling(y, x)$ on FB15K-237), inverse relations (e.g., $capitalOfArea(x, y) \leftarrow areaHasCapital(y, x)$ on FB15K-237), and relationship subsumption (e.g., $athletePlayedForTeam(x, y) \leftarrow athleteLedSportsTeam(x, y)$ on NELL-995).

In addition, for each benchmark, we checked how many of the rules produced by AnyBURL are captured by our transformation;[2] the results are shown in Table 5. For each benchmark, the first column shows the total number of rules produced by AnyBURL; the second column indicates how many of those are tree-like as per Definition 9; the third column shows how many rules from the second column are captured by our transformation as per Definition 6; and the fourth column shows the number of rules produced by AnyBURL that are captured by $T_{\mathcal{M}}$ *on the specific dataset*— that is, rules $r$ that satisfy $T_r(\mathcal{S}_I) \subseteq T_{\mathcal{M}}(\mathcal{S}_I)$. As one can see, a significant proportion of tree-like rules produced by AnyBURL are captured by $T_{\mathcal{M}}$. Additionally, a significant proportion of (not necessarily tree-shaped) rules produced by AnyBURL are captured by $T_{\mathcal{M}}$ on each specific dataset, which partially explains why many of the predictions by AnyBURL are also obtained by $T_{\mathcal{M}}$.

---

[2]We could not do this for DRUM since the version of the system available online cannot produce the rules.

## 5 RELATED WORK

*GNN-based approaches to KG completion* (Hamaguchi et al., 2017; Schlichtkrull et al., 2018; Xu et al., 2020; Teru et al., 2020; Zhang et al., 2020) are becoming increasingly popular because they are more generally applicable than earlier solutions based on graph embeddings (Rossi et al., 2021). Typically, the predictions made by these techniques cannot be fully explained symbolically.

*Rule mining* techniques for KGs fall in two categories. Systems in the first category learn a model on training data and then extract rules from it. Neural-LP (Yang et al., 2017) is an influential such approach, and it has inspired DRUM (Sadeghian et al., 2019) and Neural-Num-LP (Wang et al., 2020). Other approaches use reinforcement learning (Xiong et al., 2017; Das et al., 2018), dynamic neural module networks (Rocktäschel & Riedel, 2017; Campero et al., 2018), or graph embeddings (Omran et al., 2018; Zhang et al., 2019). However, the formal relationship between the learned model and the extracted rules is unspecified in these approaches; hence, the model and the rules can make different predictions. Systems in the second category, such as AnyBURL, use training data to identify path patterns in graphs that are represented as rules (Galárraga et al., 2015; Meilicke et al., 2019; Ahmadi et al., 2020; Gu et al., 2020); hence, they do not train an ML model.

*Inductive Logic Programming (ILP)* generates rules given a dataset and examples of positive and negative inferences so that rule application yields all positive and no negative examples (Muggleton, 1991). ILP techniques cannot typically handle noisy examples (Muggleton, 1991; Cropper & Muggleton, 2014; Si et al., 2019; Raghothaman et al., 2020). To improve resilience to noise, recent ILP systems such as $\partial$ILP (Evans & Grefenstette, 2018) interpret the ILP task as a binary classification problem and provide a differentiable implementation of deduction. These approaches compare to our work analogously to rule mining techniques. Moreover, they focus on learning rules from small datasets and generally struggle with large-scale KGs.

*Neuro-symbolic computation* aims to integrate logic with neural networks (d'Avila Garcez et al., 2002). To this end, it was shown that the immediate consequence operator of certain classes of logic programs can be approximated by recurrent Hölldobler et al. (1999), fibring Bader et al. (2005), and feed-forward networks Bader et al. (2007); however, these techniques do not show how to extract a program from a trained network. A method to extract rules captured by feed-forward networks was proposed by d'Avila Garcez et al. (2001); however, in contrast to our work, the extracted programs do not contain variables, and they are not guaranteed to be equivalent to the network. Recently, Dong et al. (2019) presented a network architecture that simulates application of function-free first-order rules to a dataset, but no algorithm for extracting rules from a trained model was provided.

Existing methods for *explaining the predictions of a GNN* focus on identifying parts of the graph in the embedding space that are most relevant to a given prediction (Ying et al., 2019; Luo et al., 2020; Lin et al., 2021). These techniques are largely independent from the specifics of the model used. In contrast, we focus on a specific type of GNN that can provide a logical proof of each prediction.

GNNs can express graph property tests such as isomorphism, existence of cliques, bipartiteness, and planarity (Xu et al., 2019; Morris et al., 2019; Garg et al., 2020). Moreover, first-order logic queries expressible by GNNs correspond to the description logic $\mathcal{ALCQ}$ (Barceló et al., 2020). Our work follows this line of research by relating GNNs and Datalog.

Similarly to our work, techniques for neural learning of monotonic functions on the ordering of real numbers (You et al., 2017; Gupta et al., 2016; Wehenkel & Louppe, 2019) also rely on nonnegative matrices and monotonic activation functions. We, however, focus on learning monotonic functions on the ordering induced by the existence of homomorphisms between datasets.

## 6 CONCLUSION AND FUTURE WORK

In this paper, we presented a GNN-based transformation of datasets that mimics a round of application of Datalog rules. The predictions made by our transformation can be explained symbolically, and we have shown our approach to be practically feasible. We see many avenues for future work. On the theoretical side, we shall develop extensions that capture more complex rules (e.g., transitivity), as well as nonmonotonic extensions such as negation or aggregation. On the practical side, we shall establish links with the existing approaches to rule mining and ILP, thus allowing us to compare the performance of all of these heterogeneous approaches in a unified way.

## ACKNOWLEDGMENTS

This work was supported by the AIDA project (Alan Turing Institute, EP/N510129/1), the SIRIUS Centre for Scalable Data Access (Research Council of Norway, project number 237889), Samsung Research UK, Siemens AG, and the EPSRC projects AnaLOG (EP/P025943/1), OASIS (EP/S032347/1) and UK FIRES (EP/S019111/1).

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

# A   PROOFS

## A.1   PROOF OF PROPOSITION 7

**Proposition 7.** *Let $T$ be an operator on datasets, let $r$ be a constant-free rule of form* (2)*, let $\mu$ be a substitution that maps each variable of $r$ to a distinct constant, and let $D_r = \{B_1\mu, \dots, B_m\mu\}$. If $T$ is monotonic under homomorphisms and $H\mu \in T(D_r)$, then $T$ captures $r$.*

*Proof.* Consider $T$, $r$, $D_r$, and $\mu$ that satisfy the prerequisites of the proposition. Moreover, consider an arbitrary dataset $D$ and an arbitrary fact $\alpha \in T_r(D)$; we show that $\alpha \in T(D)$. By the definition of operator $T_r$, there exists a substitution $\nu$ such that $\{B_1\nu, \dots, B_m\nu\} \subseteq D$ and $H\nu = \alpha$. Now let $h$ be the mapping of constants to constants such that $h(x\mu) = x\nu$ for each variable $x$ occurring in $r$. This definition of $h$ is correct since $\mu$ maps each variable in $r$ to a distinct constant. Also, rule $r$ is constant-free, so $h(D_r) = h(\{B_1\mu, \dots, B_m\mu\}) = \{B_1\nu, \dots, B_m\nu\} \subseteq D$ holds; moreover, each constant in $D_r$ is obtained by instantiating a variable of $r$, so $h$ is defined on all of $\mathsf{cn}(D_r)$. In other words, $h$ is a homomorphism from $D_r$ to $D$. Since $T$ is monotonic under homomorphisms by the assumption of the proposition, $h$ is also a homomorphism from $T(D_r)$ to $T(D)$. Moreover, $H\mu \in T(D_r)$ holds by our assumption, so the definition of homomorphisms ensures $h(H\mu) \in T(D)$. Finally, atom $H$ is constant-free, which together with the definition of $h$ ensures $h(H\mu) = H\nu$. Thus, we have $H\nu \in T(D)$, as required. $\qquad\square$

## A.2   PROOF OF THEOREM 8

We prove Theorem 8 in two steps. First, we first introduce in Definition A1 a notion of homomorphisms on $(\mathsf{Col}, \delta)$-graphs that is analogous to the notion of homomorphisms on datasets. Then, in Lemma A2 we show that the transformation of $(\mathsf{Col}, \delta)$-graphs by an MGNN preserves such homomorphisms. Theorem 8 finally combines this result with our encoding and decoding steps. To formalise these results, we extend the order $\leq$ on real numbers to a partial order on vectors so that $\mathbf{a}_1 \leq \mathbf{a}_2$ holds whenever $(\mathbf{a}_1)_i \leq (\mathbf{a}_2)_i$ holds for each position $i$.

**Definition A1.** *A* label-respecting homomorphism *from a $(\mathsf{Col}, \delta)$-graph $\langle V, \{E_c\}_{c\in\mathsf{Col}}, \lambda \rangle$ to a $(\mathsf{Col}, \delta)$-graph $\langle V', \{E'_c\}_{c\in\mathsf{Col}}, \lambda' \rangle$ is a mapping $h : V \to V'$ such that $\lambda(v) \leq \lambda'(h(v))$ for each vertex $v \in V$, and, for each colour $c \in \mathsf{Col}$ and each edge in $E_c$ between vertices $v$ and $w$, vertices $h(v)$ and $h(w)$ are connected by an edge in $E'_c$.*

**Lemma A2.** *For each $(\mathsf{Col}, \delta)$-MGNN $\mathcal{M}$ and all $(\mathsf{Col}, \delta)$-graphs $G$ and $G'$, each label-respecting homomorphism $h$ from $G$ to $G'$ is also a label-respecting homomorphism from $\mathcal{M}(G)$ to $\mathcal{M}(G')$.*

*Proof.* Consider a $(\mathsf{Col}, \delta)$-MGNN $\mathcal{M}$ with components as specified in Definition 3, $(\mathsf{Col}, \delta)$-graphs $G = \langle V, \{E_c\}_{c\in\mathsf{Col}}, \lambda \rangle$ and $G' = \langle V', \{E'_c\}_{c\in\mathsf{Col}}, \lambda' \rangle$, and a label-respecting homomorphism $h$ from $G$ to $G'$. Since $\mathcal{M}(G)$ and $\mathcal{M}(G')$ have the same edges as $G$ and $G'$, respectively, $h$ satisfies the first condition of a label-preserving homomorphism from $\mathcal{M}(G)$ to $\mathcal{M}(G')$. We show that $h$ also satisfies the second condition. To this end, let $\lambda^0, \dots, \lambda^L$ and $\lambda'^0, \dots, \lambda'^L$ be the sequences of vertex labelling functions used in the computation of $\mathcal{M}(G)$ and $\mathcal{M}(G')$, respectively. We next show by induction on $0 \leq \ell \leq L$ that $\lambda^\ell(v) \leq \lambda'^\ell(h(v))$ holds for each vertex $v \in V_1$; since classification function $\mathsf{cls}$ is monotonically increasing, this will imply that $\mathsf{cls}(\lambda^L(v)) \leq \mathsf{cls}(\lambda'^L(h(v)))$ holds for each vertex $v \in V_1$ as well, which proves our claim.

The base case for $\ell = 0$ holds immediately since $\lambda^0 = \lambda$ and $\lambda'^0 = \lambda'$. For the inductive step, we assume that, for some $\ell - 1$, we have $\lambda^{\ell-1}(v) \leq \lambda'^{\ell-1}(h(v))$ for each $v \in V_1$, and we show that $\lambda^\ell(v) \leq \lambda'^\ell(h(v))$ also holds for each $v \in V_1$. To this end, consider an arbitrary $i$ between 1 and the dimension of $\mathbf{b}^\ell$. By Definition 3, the $i$-th components of $\lambda^\ell(v)$ and $\lambda'^\ell(h(v))$ are equal to $\sigma(x_1)$ and $\sigma(x_2)$, where $x_1$ and $x_2$ are as in (3) and (4) for $(\mathbf{A}^\ell)_i$ and $(\mathbf{B}^\ell_c)_i$ the $i$-th rows of $\mathbf{A}^\ell$ and $\mathbf{B}^\ell_c$, respectively:

$$x_1 = (\mathbf{A}^\ell)_i\, \lambda^{\ell-1}(v) \qquad + \sum_{c\in\mathsf{Col}} (\mathbf{B}^\ell_c)_i \max\{\lambda^{\ell-1}(w) \mid w \in N_c(v)\} \qquad + (\mathbf{b}^\ell)_i, \qquad (3)$$

$$x_2 = (\mathbf{A}^\ell)_i\, \lambda'^{\ell-1}(h(v)) + \sum_{c\in\mathsf{Col}} (\mathbf{B}^\ell_c)_i \max\{\lambda'^{\ell-1}(u) \mid u \in N_c(h(v))\} + (\mathbf{b}^\ell)_i. \qquad (4)$$

By the inductive hypothesis, we have $\lambda^{\ell-1}(v) \leq \lambda'^{\ell-1}(h(v))$; moreover, all numbers in $\mathbf{A}^\ell$ are nonnegative, so we clearly have $(\mathbf{A}^\ell)_i \lambda^{\ell-1}(v) \leq (\mathbf{A}^\ell)_i \lambda'^{\ell-1}(h(v))$. In addition, mapping $h$ satisfies the first property of label-preserving homomorphisms so, for each colour $c \in \mathsf{Col}$ and each $w \in N_c(v)$, we have $h(w) \in N_c(h(v))$; moreover, by the inductive hypothesis, we have $\lambda^{\ell-1}(w) \leq \lambda'^{\ell-1}(h(w))$. By combining these two observations, we have

$$\max\{\lambda^{\ell-1}(w) \mid w \in N_c(v)\} \leq \max\{\lambda'^{\ell-1}(u) \mid u \in N_c(h(v))\},$$

which, together with the fact that all numbers in $(\mathbf{B}_c^\ell)_i$ are nonnegative, implies

$$(\mathbf{B}_c^\ell)_i \max\{\lambda^{\ell-1}(w) \mid w \in N_c(v)\} \leq (\mathbf{B}_c^\ell)_i \max\{\lambda'^{\ell-1}(u) \mid u \in N_c(h(v))\}.$$

Consequently, we have $x_1 \leq x_2$. Finally, activation function $\sigma$ is monotonically increasing, so $\sigma(x_1) \leq \sigma(x_2)$ holds, which completes our proof of the inductive step. $\qquad\square$

**Theorem 8.** *For each $(\mathsf{Col}, \delta)$-MGNN $\mathcal{M}$, mapping $T_\mathcal{M}$ is monotonic under homomorphisms.*

*Proof.* Consider an arbitrary $(\mathsf{Col}, \delta)$-MGNN $\mathcal{M}$, arbitrary datasets $D$ and $D'$, and an arbitrary homomorphism $h$ from $D$ to $D'$; thus, $h(D) \subseteq D'$ holds. Now let $G_D$ and $G_{D'}$ be $(\mathsf{Col}, \delta)$-graphs obtained by transforming $D$ and $D'$ as in Definition 2. Moreover, let $g$ be the mapping from the vertices of $G_1$ to the vertices of $G_2$ defined as follows:

- $g(v_a) = v_{h(a)}$ for each vertex $v_a$ of $G_1$; and
- $g(v_{a,b}) = v_{h(a),h(b)}$ for each vertex $v_{a,b}$ of $G_1$.

It is straightforward to see that $g$ is a label-respecting homomorphism from $G_D$ to $G_{D'}$. By Lemma A2, mapping $g$ is also a label-respecting homomorphism from $\mathcal{M}(G_D)$ to $\mathcal{M}(G_{D'})$. This, however, implies that $h$ is also a homomorphism from $T(D)$ to $T(D')$, as required. $\qquad\square$

### A.3 Proof of Theorem 10

**Theorem 10.** *For each $(\mathsf{Col}, \delta)$-MGNN $\mathcal{M}$ and each dataset $D$, we have $T_\mathcal{M}(D) = T_{\mathcal{P}_\mathcal{M}}(D)$.*

*Proof.* Consider arbitrary $\mathcal{M}$, $\mathcal{P}_\mathcal{M}$, and $D$ as stated in the claim. Let $L$ be the number of layers of $\mathcal{M}$, and let $k$ be the dimension of $\mathcal{M}$. Since operator $T_{\mathcal{P}_\mathcal{M}}$ captures each rule in $\mathcal{P}_\mathcal{M}$ by definition, we clearly have $T_{\mathcal{P}_\mathcal{M}}(D) \subseteq T_\mathcal{M}(D)$. For the converse, consider an arbitrary fact $\alpha \in T_\mathcal{M}(D)$. We next construct a rule $r$ for which we prove $r \in \mathcal{P}_\mathcal{M}$ and $\alpha \in T_r(D)$.

Let $G_D$ be the $(\mathsf{Col}, \delta)$-graph obtained from $D$ by the transformation in Definition 2, and let $V_D$ be the set of vertices of $G_D$. For each $\ell \in \{0, \ldots, L\}$, let $\mathbf{v}^\ell$ be the vector labelling each $v \in V_D$ in the computation of $\mathcal{M}(G_D)$. Moreover, let $\mu$ be a substitution that maps each variable to a distinct constant. We next construct an atom $H$, a conjunction $C$, a substitution $\nu$ from the variables in $C$ to $\mathsf{cn}(D)$, a set $U$ of fresh vertices (i.e., vertices not occurring in $G_D$) of the form $u_d$ and $u_{d,e}$ for $d$ and $e$ constants from the image of $\mu$, and mappings $\xi : U \to V_D$ and $M_c : U \to 2^{V_D}$ for each $c \in \mathsf{Col}$. We also assign each vertex in $U$ to a level between $0$ and $L$, and we identify a single vertex from $U$ as the *root* vertex. In the rest of this proof, we use letters $a$ and $b$ for the constants in $D$, letters $d$ and $e$ for the constants in the image of $\mu$, possibly indexed letters $v$, $w$, and $s$ for the vertices in $V_D$, and possibly indexed letters $u$ and $t$ for the vertices in $U$. Our construction is by induction from level $L$ down to level 1, and we prove in parallel the following properties:

- P1. for each vertex of the form $u_d \in U$, for $x$ the variable such that $\mu(x) = d$, and for $a = \nu(x)$, we have $\xi(u_d) = v_a$;

- P2. for each vertex of the form $u_{d,e} \in U$, for $x$ and $y$ variables such that $\mu(x) = d$ and $\mu(y) = e$, and for $a = \nu(x)$ and $b = \nu(y)$, we have $\xi(u_{d,e}) = v_{a,b}$;

- P3. conjunction $C$ satisfies $\nu(C) \subseteq D$; and

- P4. if $H$ is of the form $A(x)$, conjunction $C$ is $(3k, L)$-tree-like for $x$; otherwise, $C$ is of the form $C_x \wedge B_{x,y} \wedge C_y$, where $C_x$ and $C_y$ are $(3k, L)$-tree-like conjunctions for $x$ and $y$, respectively, and $B_{x,y}$ is a nonempty conjunction of atoms of the form $S(x, y)$ or $S(y, x)$.

We initialise $C$ as the empty conjunction, and we set $\nu$, $\xi$, and each $M_c$ for $c \in \mathsf{Col}$ as the empty mappings. For the induction base, we consider the form of the atom $\alpha \in T_{\mathcal{M}}(D)$ and proceed as follows; the base case also identifies the root vertex. Properties P1–P4 clearly hold after these steps.

- If $\alpha$ is of the form $A(a)$, then $V_D$ contains vertex $v_a$. We introduce a fresh variable $x$, and we let $d$ be the constant such that $d = \mu(x)$. We define $\nu(x) = a$; we define $H = A(x)$; we introduce vertex $u_d$ of level $L$; we define $\xi(u_d) = v_a$; and we make $u_d$ the root vertex. Finally, we extend $C$ with atom $B(x)$ for each $B(a) \in D$. Note that $C$ can be empty if $D$ does not contain a fact such as $B(a)$. However, $v_a \in V_D$ implies that constant $a$ must occur in $D$ in a fact of the form $S(a, b)$ or $S(b, a)$. Thus, $N_c(v_a) \neq \emptyset$ for $c \in \{c_1, c_2, c_4\}$. Since $L \geq 1$, vertex $u_d$ is processed in the inductive step below, which will extend $C$ with an atom that contains variable $x$.
- If $\alpha$ is of the form $R(a, b)$, then $V_D$ contains vertices $v_{a,b}$ and $v_{b,a}$; due to the latter, $D$ contains at least one fact of the form $S(a, b)$ or $S(b, a)$, and so $V_D$ contains vertices $v_a$ and $v_b$. We introduce fresh variables $x$ and $y$, and we let $d$ and $e$ be constants such that $d = \mu(x)$ and $e = \mu(y)$. We define $\nu(x) = a$ and $\nu(y) = b$; we define $H = R(x, y)$; we introduce vertices $u_d, u_e, u_{d,e}$, and $u_{e,d}$ of level $L$; we define $\xi(u_d) = v_a, \xi(u_e) = v_b, \xi(u_{d,e}) = v_{a,b}$, and $\xi(u_{e,d}) = v_{b,a}$; and we make $u_{d,e}$ the root vertex. Moreover, we extend $C$ with an atom $S(x, y)$ for each $S(a, b) \in D$, an atom $S(y, x)$ for each $S(b, a) \in D$, an atom $B(x)$ for each $B(a) \in D$, and an atom $B(y)$ for each $B(b) \in D$; note that $C$ cannot be empty after this step. Finally, we define $M_c(u) = N_c(\xi(u))$ for each $u \in \{u_{d,e}, u_{e,d}\}$ and colour $c \in \mathsf{Col}$; note that $M_{c_1}(u)$, $M_{c_2}(u)$, and $M_{c_3}(u)$ are all singletons, and $M_{c_4}(u) = \emptyset$.

Next, we consider each level $\ell$ with $1 \leq \ell \leq L$ that has already been processed, and we consider each vertex of the form $u_d$ of level $\ell$. Let $x$ be the variable such that $\mu(x) = d$, and let $a = \nu(x)$; property P1 ensures $\xi(u_d) = v_a$. For each colour $c \in \{c_1, c_2, c_4\}$, we choose a smallest set $M_c(u_d) \subseteq N_c(v_a)$ such that, for each $\ell' \in \{0, \ldots, \ell - 1\}$ and each $i \in \{1, \ldots, k\}$, there exists a vertex $w \in M_c(u_d)$ satisfying

$$(\mathbf{w}^{\ell'})_i = \max\{(\mathbf{s}^{\ell'})_i \mid s \in N_c(v_a)\}. \tag{5}$$

Note that set $M_c(u_d)$ may not be unique, but any set satisfying (5) can be selected. There are at most $k$ distinct values of $(\mathbf{w}^{\ell'})_i$ for each $\ell'$, so set $M_c(u_d)$ contains at most $k\ell$ elements. We also define $M_{c_3}(u_d) = \emptyset$. We now consider each vertex $v \in M_{c_1}(u_d) \cup M_{c_2}(u_d) \cup M_{c_4}(u_d)$. By Definition 2, such $v$ is of the form $v_{a,b}$, $v_{b,a}$, or $v_b$ for $b \in \mathsf{cn}(D)$, depending on the set from which $v$ is drawn. Set $V_D$ clearly also contains vertices $v_{b,a}$ and $v_b$ if $v \in M_{c_1}(u_d)$, and vertices $v_{a,b}$ and $v_b$ if $v \in M_{c_2}(u_d)$. Moreover, if $v \in M_{c_4}(u_d)$, then $V_D$ clearly contains $v_b$, and $M_{c_4}(u_d) \subseteq N_{c_4}(v_a)$ ensures that $D$ contains a fact of the form $S(a, b)$ or $S(b, a)$; thus, $V_D$ also contains $v_{a,b}$ and $v_{b,a}$. We introduce a fresh variable $y$, and we let $e$ be the constant such that $e = \mu(y)$. We define $\nu(y) = b$; we introduce vertices $u_e, u_{d,e}$, and $u_{e,d}$ of level $\ell - 1$; and we define $\xi(u_e) = v_b, \xi(u_{d,e}) = v_{a,b}$, and $\xi(u_{e,d}) = v_{b,a}$. Moreover, we extend $C$ with an atom $S(x, y)$ for each $S(a, b) \in D$, an atom $S(y, x)$ for each $S(b, a) \in D$, and an atom $B(y)$ for each $B(b) \in D$. Finally, we define $M_c(u) = N_c(\xi(u))$ for each $u \in \{u_{d,e}, u_{e,d}\}$ and colour $c \in \mathsf{Col}$; note that $M_{c_1}(u)$, $M_{c_2}(u)$, and $M_{c_3}(u)$ are all singletons, and $M_{c_4}(u) = \emptyset$. Since each $M_c(u_d)$ with $c \in \{c_1, c_2, c_4\}$ contains at most $k\ell$ vertices and we consider three colours, this step introduces at most $3k\ell$ fresh variables. Moreover, note that each fresh variable $y$ occurs in $C$ in an atom of the form $S(x, y)$ or $S(y, x)$. Finally, it should be clear that P1–P4 hold after level $\ell$ is processed.

This completes our inductive construction. At this point, either $H = A(x)$ and $C$ is a nonempty $(3k, L)$-tree-like conjunction for $x$, or $H = R(x, y)$ and $C = C_x \wedge B_{x,y} \wedge C_y$ where $C_x$ and $C_y$ are $(3k, L)$-tree-like conjunctions for $x$ and $y$, respectively, and $B_{x,y}$ is a nonempty conjunction of atoms of the form $S(x, y)$ or $S(y, x)$. Thus, rule $H \leftarrow C$ is of the form $A(x) \leftarrow C_x$ or $R(x, y) \leftarrow C_x \wedge B_{x,y} \wedge C_y$. To complete the proof, we next show that this rule is captured by $T_{\mathcal{M}}$. Then, Definition 9 ensures $H \leftarrow C \in \mathcal{P}_{\mathcal{M}}$; together with $H\nu = \alpha$ and $\nu(C) \subseteq D$, we have $\alpha \in T_{\mathcal{P}_{\mathcal{M}}}(D)$, which proves our claim.

We use Proposition 7 to show that $H \leftarrow C$ is captured by $T_{\mathcal{M}}$. To this end, let $D_C = \{\alpha\mu \mid \alpha \in C\}$, and let $G_{D_C}$ be the $(\mathsf{Col}, \delta)$-graph constructed using vertices in $U$ obtained by applying the encoding from Definition 2 to dataset $D_C$. Finally, let $\mathbf{u}^\ell$ be the labelling associated with each vertex $u \in U$ in the computation of $\mathcal{M}(G_{D_C})$. The construction of $C$ clearly ensures the following two properties for each colour $c \in \mathsf{Col}$ and each vertex $u \in U$:

---

**Algorithm 1** Rule extraction algorithm

> **Input:**   $m$     : natural number (max. num. of atoms in the bodies of extracted rules)
>           $\mathcal{M}$    : an MGNN

1: $head\_to\_bodies := \emptyset$
2: **for** $i \leq m$ **do**
3:     **for** $pred\_list \in \text{COMBINATIONS}(i)$ **do**
4:         **for** $body \in \text{BODIES}(pred\_list)$ **do**
5:             **if** $\text{TREELIKE}(body)$ **then**
6:                 $input\_graph := \text{ENCODE}(body)$
7:                 $output\_graph := \mathcal{M}(input\_graph)$
8:                 $heads := \text{DECODE}(output\_graph)$
9:                 **for** $head \in heads$ **do**
10:                     $new\_head, new\_body := \text{NORMALISE}(head, body)$
11:                     $not\_implied := true$
12:                     **for** $body' \in head\_to\_bodies[new\_head]$ **do**
13:                         **if** $\text{IMPLIES}(body', new\_body)$ **then** $not\_implied := false$
14:                     **if** $not\_implied$ **then** $\text{ADD}(head\_to\_bodies[new\_head], new\_body)$
15: $extracted\_rules := \emptyset$
16: **for** $\langle head, bodies \rangle \in head\_to\_bodies$ **do**
17:     **for** $body \in bodies$ **do**
18:         $\text{ADD}(extracted\_rules, head \leftarrow body)$
    **return** $extracted\_rules$

---

> R1.  for each vertex $v \in M_c(u)$, there exists a vertex $t \in N_c(u)$ such that $\xi(t) = v$; and

> R2.  for each vertex $t \in N_c(u)$ in graph $G_{D_C}$, we have $\xi(t) \in N_c(\xi(u))$.

We now show by the induction on $\ell$ that, for each $0 \leq \ell \leq L$ and each vertex $u \in U$ whose level is at least $\ell$, we have $\mathbf{u}^\ell = \mathbf{v}^\ell$ for $v = \xi(u)$. For the base case $\ell = 0$, we have the following possibilities.

- Assume $u$ is of the form $u_d$. Let $x$ be the variable such that $d = \mu(x)$, and let $a = \nu(x)$. Property P1 ensures $\xi(u_d) = v_a$, and the construction of $C$ ensures $B(x) \in C$ if and only if $B(a) \in D$; thus, $\mathbf{u}_d^0 = \mathbf{v}_a^0$ holds.
- Assume $u$ is of the form $u_{d,e}$. Let $x$ and $y$ be variables such that $\mu(x) = d$ and $\mu(y) = e$, and let $a = \nu(x)$ and $b = \nu(y)$. Property P2 ensures $\xi(u_{d,e}) = v_{a,b}$, and the construction of $C$ ensures $S(x, y) \in C$ if and only if $S(a, b) \in D$; thus, $\mathbf{u}_{d,e}^0 = \mathbf{v}_{a,b}^0$ holds.

For the induction step, assume that the property holds for some $\ell - 1$, and consider an arbitrary vertex $u \in U$ whose level is at least $\ell$. Let $v = \xi(u)$. The induction assumption ensures $\mathbf{u}^{\ell-1} = \mathbf{v}^{\ell-1}$. Also, for each colour $c \in \text{Col}$, our construction ensures that each vertex $t \in N_c(u)$ is of level at least $\ell - 1$, so the induction assumption ensures $\mathbf{t}^{\ell-1} = \mathbf{w}^{\ell-1}$ for $w = \xi(t)$. Finally, property R2 ensures that each $\mathbf{t}^{\ell-1}$ used to compute $\mathbf{u}^\ell$ is considered when computing $\mathbf{v}^\ell$, and property R1 ensures that the maximal values of $\mathbf{t}^{\ell-1}$ are also present in the computation of $\mathbf{u}^\ell$. Thus, $\mathbf{u}^\ell = \mathbf{v}^\ell$ holds.

To complete the proof, let $u$ be the root vertex, and let $v = \xi(u)$. The above property ensures $\mathbf{u}^L = \mathbf{v}^L$, and so $\text{cls}(\mathbf{u}^L) = \text{cls}(\mathbf{v}^L)$ as well. Thus, we have $\mu(H) \in T_\mathcal{M}(D_C)$, so Proposition 7 allows us to conclude that rule $H \leftarrow C$ is captured by $T_\mathcal{M}$.     □

## B   OPTIMISED RULE EXTRACTION ALGORITHM

We now describe the rule extraction algorithm that we used to obtain the results shown in Table 4 of Section 4. Our algorithm takes as input a positive integer $m$ and an MGNN $\mathcal{M}$ of dimension $k$ and with $L$ layers, and it produces a set of $(3k, L)$-tree-like rules with at most $m$ body atoms that are captured by $T_\mathcal{M}$. The algorithm ensures that no rule that is logically implied by any other individual rule is included in the output. The algorithm's pseudocode is shown in Algorithm 1, and it uses several auxiliary functions that will be introduced shortly.

The algorithm initialises in line 1 an empty mapping $head\_to\_bodies$ that will be used to store pairs of the form $\langle S(x,y), \{C_1, C_2, \ldots, C_n\}\rangle$ such that, for each $C_i$, rule $S(x,y) \leftarrow C_i$ is $(3k, L)$-tree-like, has at most $m$ body atoms, and is captured by $T_{\mathcal{M}}$. In each iteration $i$ of the main loop (lines 2–14), the algorithm produces rules with exactly $i$ body atoms. To this end, the algorithm calls COMBINATIONS$(i)$ in line 3 to produce all ordered combinations (with repetition) of predicates of length $i$. For each list of predicates $pred\_list$, the algorithm calls BODIES$(pred\_list)$ in line 4 to enumerate every possibly body over the predicates of $pred\_list$. For example, if $pred\_list$ is $[A, R]$, where $A$ is unary and $R$ is binary, then BODIES$(pred\_list)$ returns

$$\{ A(x) \wedge R(x,x), \ A(x) \wedge R(x,y), \ A(x) \wedge R(y,x), \ A(x) \wedge R(y,y) \}.$$

Next, the algorithm filters out bodies that are not $(3k, L)$-tree-like (line 5), calls ENCODE$(body)$ to transform $body$ into a $(\mathsf{Col}, \delta)$-graph by replacing each variable by a distinct constant and then applying the encoding from Definition 2 (line 6), applies $\mathcal{M}$ to the resulting graph (line 7), and finally calls DECODE$(output\_graph)$ to reverse the encoding and obtain the set $heads$ of candidate head atoms (line 8). Thus, for each $head \in heads$, operator $T_{\mathcal{M}}$ captures $head \leftarrow body$. For each candidate $head \in heads$, the algorithm calls NORMALISE$(head, body)$ to rewrite the candidate rule $head \leftarrow body$ into $new\_head \leftarrow new\_body$ to ensure that the head of each extracted rule always uses variables $x$ and $y$ in this order (the body is rewritten accordingly). The algorithm then checks whether rule $new\_head \leftarrow new\_body$ is implied by any rule previously stored in $head\_to\_bodies$ (lines 12–13); if not, it records the new rule by adding conjunction $new\_body$ to the set $head\_to\_bodies(new\_head)$ (line 14). Finally, the algorithm transforms the mapping $head\_to\_bodies$ of heads to bodies into standard rules (lines 16–18).

Algorithm 1 satisfies the following property, which ensures its correctness.

**Proposition B1.** *Let $\mathcal{P}$ be the result of applying Algorithm 1 to an integer $m \in \mathbb{N}$ and a $(\mathsf{Col}, \delta)$-MGNN $\mathcal{M}$ of dimension $k$ and $L$ layers.*

1. *Each rule $r \in \mathcal{P}$ is $(3k, L)$-tree-like with at most $m$ body atoms, and it is captured by $T_{\mathcal{M}}$.*

2. *For each $(3k, L)$-tree-like rule $r$ with at most $m$ body atoms that is captured by $T_{\mathcal{M}}$, there exists a rule in $\mathcal{P}$ that logically implies $r$.*

*Proof.* The first claim is a direct consequence of Proposition 7 and the fact that Algorithm 1 only considers $(3k, L)$-tree-like rules with at most $m$ atoms in the body. For the second claim, if $r$ is a $(3k, L)$-tree-like rule with at most $m$ body atoms, then lines 2–8 of Algorithm 1 check whether this rule is captured by $T_{\mathcal{M}}$; if that is the case, then lines 9–14 ensure that $r$ is either in the output of the algorithm, or the algorithm's output contains a rule that logically implies $r$. $\qquad\square$

## C   FULL RESULTS FOR THE RULE EXTRACTION EXPERIMENT

As we explained in Section 4, for each benchmark, we used Algorithm 1 with $m = 2$ to extract the subset $\mathcal{P} \subseteq \mathcal{P}_{\mathcal{M}}$ of all nonredundant rules with at most two body atoms, and then we applied $\mathcal{P}$ to the testing dataset $\mathcal{S}_I$ and computed the usual classification metrics. Complete results of this experiment are shown in Table 6.

| Benchmark | | Precision | Recall | Accuracy | F1 Score |
|---|---|---|---|---|---|
| FB15K-237 | v1 | 98.5 | 32.7 | 66.1 | 49.1 |
| | v2 | 100.0 | 38.3 | 69.1 | 55.4 |
| | v3 | 100.0 | 33.1 | 66.5 | 49.7 |
| | v4 | 99.6 | 32.7 | 66.3 | 49.3 |
| NELL-995 | v1 | 96.4 | 80.0 | 88.5 | 87.4 |
| | v2 | 100.0 | 45.2 | 72.6 | 62.2 |
| | v3 | 100.0 | 51.1 | 75.5 | 67.6 |
| | v4 | 99.7 | 53.9 | 76.9 | 70.0 |
| WN18RR | v1 | 100.0 | 61.7 | 80.9 | 76.3 |
| | v2 | 100.0 | 60.3 | 80.2 | 75.2 |
| | v3 | 100.0 | 27.9 | 64.0 | 43.7 |
| | v4 | 100.0 | 57.6 | 78.8 | 73.1 |

Table 6: Classification metrics for the programs extracted using Algorithm 1 with $m = 2$

