# OpenReview forum: "Explainable GNN-Based Models over Knowledge Graphs"
_ICLR.cc/2022/Conference — ICLR 2022 Poster_

### Official Review · Reviewer_Eofg · 2021-10-27

**Correctness:** 4
**Technical Novelty And Significance:** 3
**Empirical Novelty And Significance:** 2
**Recommendation:** 8
**Confidence:** 3

**Details Of Ethics Concerns:**

I do not think that the core methods described here are inherently problematic. However, the authors motivate their approach based on how it can explain decisions made by a machine learning system, and make a particular reference to credit applications (on page 2, reproduced below):

> For example, we could train an MGNN on a graph containing examples of credit applications and then decide new applications by applying the model to a different graph; this is analogous to how deep learning is already used in practice. However, unlike the existing approaches, we can explicate the rules that are implicitly captured by the model and use them to fully explain each credit decision.

Here the authors seem to be working under the assumption that having the classification come from a collection of Datalog rules makes it a "better" choice than using an ordinary GNN for a credit application decision. But as I discuss in my main review, I don't think their method necessarily extracts a good explanation for the behavior of the system. Furthermore, I don't think that being able to explain the conclusions in this manner is sufficient for addressing the complex societal issues for a decision such as credit application.

I'm not sure that this warrants a full ethics review, but I am flagging it because it stood out to me (and because the authors do not include any ethics discussion in the paper). I think it would be good for the authors to expand on the ethical implications of their approach if applied to situations like this, and in particular on how an explanation in terms of Datalog rules might be used in practice. (For instance, is the idea that a human would inspect the generated rules for a particular graph to resolve a credit application dispute? Or inspect them in advance to determine whether the model is doing something unfair? Are either of these actually feasible for the rulesets generated by the proposed method, given the extremely large number of rules?)

**Update (Nov 22):** My concerns here have been mostly addressed by the new discussion of how to use the extracted Datalog rules. It might still be worthwhile to add an ethics discussion describing the remaining limitations of the approach when used in this kind of setting.

**Main Review:**

### Clarity and technical correctness
Overall, the paper is well written. I was able to follow the majority of the main paper, and the authors anticipated many of my technical questions and answered them with helpful examples and discussion (for instance, by walking through the intuitive interpretation of MGNNs, discussing how their method does not allow inferring transitivity relations, and describing how they set thresholds in their experiments).

Some specific comments on clarity:
- On page 5, it was difficult to follow the connection between the definition of "monotonic under homomorphisms" and the description of "monotonic" for constant-free programs; the paper simply states "Monotonicity under homomorphisms combines both of these properties." without saying how. (Is it that we can always construct a homomorphism from a smaller dataset to a larger one, and thus properties of the smaller dataset must also hold in the larger one?) It would be nice to have a short discussion of why an operator that is monotonic under homomorphisms is always monotonic.
- It would be useful to have a figure showing what the graph encoding of a dataset looks like.
- In section 4, it says the MGNN models were trained with "cross-entropy loss". What was used as the output probability for this loss? Was it a logistic sigmoid applied to $\lambda^L(v)$? This doesn't appear to be stated explicitly.

I believe the theorems are likely to be true, but I found some of the proofs difficult to follow and did not check them in detail. Although the paper does a good job at giving intuition in the main paper, I felt like this intuition was missing in some of the proofs, in particular in Theorem 10. Some things I had trouble following for the proof of Theorem 10:
- There are a huge number of definitions and new symbols being introduced, and I wasn't sure what those symbols were supposed to represent.
- The induction argument is very difficult to follow, I think because it omits the dependence of various quantities on the induction variable $\ell$ (I think), interleaves re-definition of values with proofs about them, and doesn't seem to have a well-stated goal (or at least I couldn't find it). As one point of confusion, after the end of the construction, it is stated that "C is a nonempty (3k, L)-tree-like conjunction for x" but I don't think this is ever explicitly proved, or even stated as something that should be proved; it seems like this should have been a property being maintained through the induction process?

### Interpretation of the extracted "explanations"
**(Edit: My original criticisms in this section are partially based on a flawed comparison to an alternative approach that I later realized did not make sense. I have added strikethrough for the parts I no longer believe to be correct.)**

The authors motivate their approach based on the quality of the explanations it extracts, and one of the stated strengths of their method is that the behavior of the MGNN can be fully characterized by a set of Datalog rules. The authors seem to be working under the assumption that Datalog rules are inherently better explanations than the original GNN. While this is plausible for hand-written Datalog rules, I'm not at all convinced the rules extracted by their method are particularly useful.

From what I could understand, it seems like their construction is roughly as follows:
- Constrain an MGNN such that it is monotonic under homomorphisms.
- Enumerate *all possible rules* up to a given size (under some constraints), where the size is determined by the receptive field of the MGNN. For each rule, construct a subgraph representing the body of the rule, and see if the MGNN classifies the head of the rule as true or false.
- ~~Collect *every subgraph* (or, equivalently, every rule) that the MGNN classifies as true and store them in a gigantic list.~~ **Edit: It's not possible to enumerate every possible subgraph of every possible dataset, since the MGNN may have arbitrarily many nodes in its receptive field. The authors describe a particular subset of rules, and concrete (sub)graphs that determine whether the GNN captures those rules. This subset is smaller than the set of all subgraphs (in particular it's finite). Additionally, the revised paper clarifies that they only include rules that are not subsumed by other rules, so every part of the extracted rules is actually relevant to the MGNN output.**
- ~~Use this list (effectively a lookup table) as an explanation of the MGNN.~~ **Edit: This is an oversimplification on my part; given that they restrict the form of considered subgraphs and do subsumption checks, it's not just a lookup table.**

The argument of Theorem 10, as far as I could follow it, appears to be that if the MGNN deduces that a particular fact as true in a knowledge graph D, then the subgraph of D that the MGNN looked at could be represented as a rule, and so it would be in the gigantic list of all possible subgraphs that the MGNN classifies as true.

~~While it is technically true that a gigantic list of all possible subgraphs that the MGNN classifies as true "fully characterizes" the set of graphs that the MGNN classifies as true, this doesn't really seem like a satisfactory explanation for why it does so. In fact, I think you could do a very similar thing for an ordinary GNN without monotonicity constraints: enumerate all possible subgraphs up to the receptive field of the GNN, and see which ones it classifies as true. But such an explanation seems mostly useless, since it boils down to a sort of tautology: "the output was true because this is one of the graphs in the list of things classified as true". Admittedly, it might be possible to simplify the MGNN explanation by only keeping the set of "minimal subgraphs" classified as true by the MGNN, and discarding any rule that is implied by another smaller rule (instead of using every extracted rule as their theorem assumes). It's also possible that the constraints on the rules lead to a smaller number of subgraphs (although it still seems like an exponentially large blowup). However, it's not obvious to me that this necessarily leads to set that is small enough to reason about.~~ **Edit: This comparison is flawed; it's not just a subgraph search, because there are infinitely many possible subgraphs (among the set of all datasets) and you can't enumerate all of them. The authors also do subsumption checks to ensure that the extracted rules are as simple as possible.**

Perhaps the authors can provide additional information about how many rules are needed to actually fully characterize their models? On page 8 they say "To reduce the search space, we considered only rules with at most two body atoms", which might suggest that it is computationally infeasible to actually fully characterize the performance of MGNNs by extracting rules of this form.

Additionally, it's possible that I am misunderstanding the rule extraction mechanism; if so, please let me know.

### Novelty and related work
I am not well versed in the knowledge base completion literature, but as far as I know the idea of constraining a GNN to be monotonic under homomorphisms of knowledge bases, and thus to behave like Datalog rules, is novel. However, there has been a fair amount of related work on neural networks that are monotonic for other purposes (and which use similar techniques of nonnegative matrices and monotonic activation functions, among others). As a few examples:

- You, Seungil, et al. "Deep lattice networks and partial monotonic functions." Proceedings of the 31st International Conference on Neural Information Processing Systems. 2017. https://arxiv.org/abs/1709.06680
- Gupta, Maya, et al. "Monotonic calibrated interpolated look-up tables." The Journal of Machine Learning Research 17.1 (2016): 3790-3836. https://arxiv.org/abs/1505.06378
- Wehenkel, Antoine, and Gilles Louppe. "Unconstrained monotonic neural networks." Advances in Neural Information Processing Systems 32 (2019): 1545-1555. https://arxiv.org/abs/1908.05164

The technical details of the rule extraction mechanism seem novel as well, but as I've stated above, I'm not particularly convinced that the rules it extracts are particularly meaningful, so it's difficult to tell if this is significant.

### Empirical results
The authors evaluate their approach compared to two existing knowledge graph completion baselines. From the results, it seems like their method is not strictly better or worse than either of the baselines they present, and the different methods have a wide range of performance across the different datasets. One thing that would be useful would be to report standard deviation measures over different random seeds and training splits (specifically the 9:1 split used to construct the incomplete knowledge base and completion targets).

The authors note that the expressivity of MGNNs is limited by the power of "(c,d)-tree-like" rules, and that consequently they cannot express properties such as transitivity. They point to this as a possible reason for the low recall of their method on many of the datasets. They also include an experiment with an alternative test set where they ensure every output fact is at least structurally possible to be classified using their method, and show that recall improves; however, they don't report baseline performance for this test set so it's possible some of the improvement is just because this new test set is easier.

The authors also mention that they analyzed some of the rules by hand. I think it would be useful to include additional details of this, for instance by showing some of the extracted rules. I also wonder how close their extracted rules come to capturing the behavior of the MGNN; how many classifications of the MGNN can be explained using the extracted rules, and how many of them are left unexplained due to the truncation on the search method?

I also notice that their method learns orders of magnitude more rules than AnyBURL, e.g. their method learns roughly 90,000 rules on FB15K-237 v4, whereas AnyBURL achieves higher accuracy with only 3,574 rules. It seems like fewer rules should be better for a good explanation, right?

### Updated review (Nov 18)

After discussion with the authors, I have raised my score from 5 to 6. I misunderstood a few aspects of the proposed method in my initial review, and also compared it to an alternative approach which I later realized was incorrect. The authors have also addressed many of my concerns regarding clarity and missing experimental details.

I still feel that the paper could be improved by adding more of a discussion in the main paper of how to actually make use of the extracted rules to explain predictions made by a MGNN in practice. For instance, given the extracted set of rules, how do we formally verify properties of interest about the MGNN? Or, given a particular prediction of the MGNN in a particular dataset, how can we identify which facts the MGNN actually used to make its prediction? Can the full set of rules (e.g. those with more than two body atoms) be extracted quickly enough to make either of these practical? These details are especially important because the paper motivates the approach based on the trustworthiness/compliance/verifiability/fairness of the system. After discussion with the authors I believe doing this kind of verification and inspection should be possible in principle (although I have some concerns about runtime), but many of the details are buried in the appendix, and I think they should be given more of a focus.

### Updated review (Nov 22)

I have raised my score again from 6 to 8.
- The authors have added additional information about how to use Datalog reasoning engines to explain the behavior of the MGNN using the extracted rules.
- I am still unsure that it is feasible to extract every rule captured by the MGNN, which would make verification difficult. However, the authors have added new results showing that the subset of extracted rules with at most two body atoms still have fairly similar performance to the MGNN on their datasets. If verification is important, a user could then just use the extracted rules instead of the MGNN, allowing them to build a fully-explainable prediction system while trading off accuracy and computational cost (e.g. spending compute to extract more rules in exchange for a better, more-accurate approximation of the MGNN).

A few more suggestions for improvement:
- I would recommend briefly mentioning the computational constraints on the current rule-extraction algorithm in the introduction, to make it clear that there is a difference between the theoretical and practical ability to extract rules. Specifically, after the sentence "we can explicate the rules that are implicitly captured by the model and use them to fully explain each credit recommendation", I'd suggest adding another sentence stating that the existing algorithm for extracting rules may only explain a subset of predictions of the original MGNN (and, optionally, that the extracted rules could just be used directly if it is critical to be able to explain every prediction).
- In section 4 "Rule Extraction" after "we applied $T_P$ to the test dataset of each benchmark", it would be useful to explicitly refer to Table 6 in the appendix, so that readers know where to find that information.
- I think it would be important to include some results for AnyBURL and DRUM in Table 6 as well, by extracting a set of rules from them and then evaluating those rules. I am aware neither AnyBURL or DRUM has any theoretical guarantees about extracted rules, but as far as I understand it each method does produce some rules with thresholds, so it seems fair to take a subset of their rules (with threshold similarly tuned for accuracy). This would help answer the question of how much the MGNN adds over the baselines if you require that every prediction be explainable by rules.

**Summary Of The Paper:**

This paper focuses on graphs representing sets of facts in a knowledge base, and casts knowledge graph completion as a graph labeling problem. The authors propose a particular encoding of facts into graphs, and introduce a particular family of "monotonic graph neural networks" (MGNNs) that share properties with logical inference rules in Datalog: namely, they prove that MGNNs are "monotonic under homomorphisms", which effectively means that renaming objects or introducing new facts never causes the classification to switch from True to False. The authors also prove that for any MGNN (with a fixed classification threshold) there exists a set of rules in Datalog that produce the same conclusions, and treat this as a symbolic explanation for the MGNN's behavior. They evaluate MGNNs on a set of knowledge graph completion tasks, and find that it performs on par with some other methods for knowledge graph completion.

**Summary Of The Review:**

The main paper is well written and I think it is technically sound (although I found the proofs difficult to follow and did not fully verify them). ~~However, I'm not convinced that the rules they extract really count as "explanations": as far as I can tell the extracted rule sets can be extremely complex and may not have any more explanatory power than a gigantic lookup table~~. Empirical results are also not particularly strong; their system achieves comparable performance to other methods but seems to require orders of magnitude more rules and also has fundamental limits to its expressivity.

**Update (Nov 18):** After discussion with the authors, I have raised my score from 5 to 6; some of my concerns in my initial review were based on a misunderstanding of the proposed method, and the authors have addressed many of my comments regarding clarity and experimental details in the revised paper. I still feel that the paper could be improved by describing in more detail how to use the extracted rules to explain predictions, since this seems to be the main way their approach differs from prior work.

**Update (Nov 22):** I have raised my score again from 6 to 8. The authors have added more discussion of how Datalog engines can use the extracted rules and how accurately the truncated rule-extraction algorithm approximates the MGNN. Since the rule-based approximation performs fairly well (and could be improved by running the extraction algorithm for a longer time), it could be used directly for making predictions in domains where explaining every prediction is critical, and then standard Datalog tools could be used to explain all predictions.

---

> ### Author Response · Authors · 2021-11-16
> **Response to reviewer Eofg - part 1 of 4**
>
> We thank the reviewer for their
> valuable comments.
> We next discuss in detail how we have
> addressed them, and
> answer the questions raised in their review.
> We release a revised version of the paper
> in a few days, which includes all changes discussed below.
>
>  &nbsp;
> >On page 5, it was difficult to follow the connection between the definition of "monotonic under homomorphisms" and the description of "monotonic" for constant-free programs; the paper simply states "Monotonicity under homomorphisms combines both of these properties." without saying how. (Is it that we can always construct a homomorphism from a smaller dataset to a larger one, and thus properties of the smaller dataset must also hold in the larger one?) It would be nice to have a short discussion of why an operator that is monotonic under homomorphisms is always monotonic.
>
>
> The argument suggested by the reviewer is correct. The identity mapping is a homomorphism. Therefore, if $T$ is monotonic
> under homomorphisms, Definition 5 ensures that for any two datasets $D$ and $D'$ such that $D \subseteq D'$, we have $T(D) \subseteq T(D')$, so $T$ is monotonic.
>
>
>  &nbsp;
> >It would be useful to have a figure showing what the graph encoding of a dataset looks like.
>
> We add this to the revised version of the paper.
>
>  &nbsp;
> >In section 4, it says the MGNN models were trained with "cross-entropy loss". What was used as the output probability for this loss? Was it a logistic sigmoid applied to $\lambda^L(v)$?  This doesn't appear to be stated explicitly.
>
> We add this information in the revised version of the paper.
>
>  &nbsp;
> >I believe the theorems are likely to be true, but I found some of the proofs difficult to follow and did not check them in detail. Although the paper does a good job at giving intuition in the main paper, I felt like this intuition was missing in some of the proofs, in particular in Theorem 10. Some things I had trouble following for the proof of Theorem 10.
> There are a huge number of definitions and new symbols being introduced, and I wasn't sure what those symbols were supposed to represent.
> >The induction argument is very difficult to follow, I think because it omits the dependence of various quantities on the induction variable $\ell$
> (I think), interleaves re-definition of values with proofs about them, and doesn't seem to have a well-stated goal (or at least I couldn't find it). As one point of confusion, after the end of the construction, it is stated that "C is a nonempty (3k, L)-tree-like conjunction for x" but I don't think this is ever explicitly proved, or even stated as something that should be proved; it seems like this should have been a property being maintained through the induction process?
>
>
> The proof of Theorem 10 relies on a rather involved
> mutual induction argument, and we find that interleaving re-definitions of values
> with proofs about them makes the proof cleaner and more understandable.
> We do agree, however, that it would be helpful to state explicitly the induction invariant that $C$ is a $(3k,L)$-tree-like conjunction.
> We have updated the proof with these additional details in the revised version of the paper.
> Please note that we already show that $C$ is non-empty after level $L$ is processed.

---

> ### Author Response · Authors · 2021-11-16
> **Response to reviewer Eofg - part 2 of 4**
>
> _(this is a continuation of our response to the review)_
>
> &nbsp;
> >From what I could understand, it seems like their construction is roughly as follows:
> >* Constrain an MGNN such that it is monotonic under homomorphisms.
> >* Enumerate all possible rules up to a given size (under some constraints), where the size is determined by the receptive field of the MGNN.
> >* For each rule, construct a subgraph representing the body of the rule, and see if the MGNN classifies the head of the rule as true or false.
> >* Collect every subgraph (or, equivalently, every rule) that the MGNN classifies as true and store them in a gigantic list.
> >* Use this list (effectively a lookup table) as an explanation of the MGNN.
> >* The argument of Theorem 10, as far as I could follow it, appears to be that if the MGNN deduces that a particular fact as true in a knowledge graph D, then the subgraph of D that the MGNN looked at could be represented as a rule, and so it would be in the gigantic list of all possible subgraphs that the MGNN classifies as true.
>
> >While it is technically true that a gigantic list of all possible subgraphs that the MGNN classifies as true "fully characterizes" the set of graphs that the MGNN classifies as true, this doesn't really seem like a satisfactory explanation for why it does so. In fact, I think you could do a very similar thing for an ordinary GNN without monotonicity constraints: enumerate all possible subgraphs up to the receptive field of the GNN, and see which ones it classifies as true. But such an explanation seems mostly useless, since it boils down to a sort of tautology: "the output was true because this is one of the graphs in the list of things classified as true".
>
> This summary overlooks a key property of our approach: we extract a set of rules which derives the same facts as the MGNN-based model
> *on every dataset* (note that, for a fixed signature of unary and binary predicates there are *infinitely many* different datasets because
> the number of constants that can be used in facts is unbounded). This is a much stronger result than being able to extract a set of rules
> deriving the same facts as an MGNN *on a specific dataset*, which can indeed be
> accomplished by constructing a look-up table in the way suggested by the reviewer.
> We kindly ask the reviewer to also take into account the
> following considerations:
>
> *1. The rule extraction procedure is independent of the input dataset.
> According to the reviewer,
> >the argument of Theorem 10, as far as I could follow it, appears to be that if the MGNN deduces that a particular fact as true in a knowledge graph $D$, then the subgraph of $D$ that the MGNN looked at
> could be represented as a rule
>
> this is incorrect since the argument does not show that a subgraph of $D$ can be represented as a rule, but rather that *there exists* a rule captured by the model (independently of $D$) which applied to a subgraph of $D$ derives the corresponding fact.
>
> *2. In relation to the previous point, the summary contains the following instruction: ''collect every subgraph (equivalently, every rule)'', and seemingly treats subgraphs and rules as equivalent. Please note that subgraphs and bodies of rules are very different types of objects. In particular, bodies of rules contain variables, while subgraphs do not.
> Rule bodies can match to many different subgraphs.
>
> *3. The summary states that ''one could do a very similar thing for an ordinary GNN without monotonicity constraints''. This is not the case, since GNNs are not monotonic under homomorphisms in general.
> One can easily produce a GNN that behaves
> non-monotonically, and therefore cannot be equivalent to *any* set of first-order rules since the entailment relation in
> first-order logic is monotonic under homomorphisms.
>
>
> For example, fix a list with two single unary predicates $A_1$ and $A_2$ and no binary predicates, and let $\mathcal{M}$ be the (non-monotonic) $(\textsf{Col},2)$-GNN with $1$ layer (and dimension 2) with $$\mathbf{A}^1 = \begin{pmatrix} 0 & 0 \\\\ 0 & -1 \end{pmatrix},$$ all matrices $\mathbf{B}^1$ with all coefficients equal to $0$, and bias vector $$\mathbf{b}^1 = \begin{pmatrix} 0 \\\\  1 \end{pmatrix},$$ with activation function $\sigma(x) =1$ if $x \geq 1$ and $\sigma(x) = 0$ otherwise, and $\textsf{cls}(x) = \sigma(x)$. Clearly, on input dataset $D =\\{A_1(a)\\}$, we have $T_{\mathcal{M}}(D) = \\{A_2(a)\\}$. However, on input $D' =\\{A_1(a),A_2(a)\\}$, we have $T_{\mathcal{M}}(D') = \emptyset $, even though $D \subseteq D'$. Thus
> $T_{\mathcal{M}}$ is not monotonic (and therefore it is not monotonic under homomorphisms).

---

> > ### Comment · Reviewer_Eofg · 2021-11-17
> > **Discussion of part 2 of response**
> >
> >
> > > This summary overlooks a key property of our approach: we extract a set of rules which derives the same facts as the MGNN-based model on every dataset ... This is a much stronger result than being able to extract a set of rules deriving the same facts as an MGNN on a specific dataset, which can indeed be accomplished by constructing a look-up table in the way suggested by the reviewer.
> >
> > Thank you for your clarifications. I have realized that I made an error when imagining the lookup-table alternative approach, and I think my suggestion does not actually make sense. My mistake was thinking that the number of nodes observed by a GNN with a finite number of layers is always bounded, whereas it is really only the diameter of the subgraph that is bounded, and the graph could have arbitrarily many nodes within a finite number of hops away.
> >
> > > the argument does not show that a subgraph of $D$ can be represented as a rule, but rather that there exists a rule captured by the model (independently of $D$) which applied to a subgraph of $D$ derives the corresponding fact.
> >
> > Isn't it true that the rule that is shown to exist is "designed" to exactly match the subgraph of $D$ that the GNN observed? It's perhaps an oversimplification on my part to say that the rule "represents" the subgraph, but as far as I understood it, the construction seems to say that the rule only applies to subgraphs (of perhaps some other dataset $D'$) that are isomorphic to one we found in $D$.
> >
> > (One thing I didn't realize in my initial review, though, is that Theorem 10 always identifies a subgraph of fixed maximum size that characterizes the behavior, despite the fact that the GNN might have seen arbitrarily many nodes. This does seem like a useful property that ordinary GNNs don't necessarily have.)
> >
> > > Please note that subgraphs and bodies of rules are very different types of objects. In particular, bodies of rules contain variables, while subgraphs do not.
> >
> > This may just be a notation issue. I understand that rules are not identical to particular subsets of a specific dataset, but I'm using "subgraph" loosely to refer to a configuration of labeled nodes and edges. So when I said "collect every subgraph" I meant "enumerate every configuration of labeled nodes and edges up to the receptive field of the GNN". Perhaps a more precise way to express what I mean would be to say "for every possible subgraph (up to a fixed size) of every possible dataset, up to isomorphism between subgraphs, construct a rule that applies if and only if the dataset graph contains a subgraph isomorphic to this one". Although thinking about it again, I guess perhaps "isomorphic" is too strong, and it would be better to say "homomorphic" here.
> >
> > > One can easily produce a GNN that behaves non-monotonically, and therefore cannot be equivalent to any set of first-order rules since the entailment relation in first-order logic is monotonic under homomorphisms.
> >
> > Sorry, by "similar thing" I didn't mean "extract a set of first-order rules", but rather just "extract a similarly-sized description of the behavior of the system that provably behaves identical to the GNN". But I realize now that an explanation based on subgraphs without using the monotonicity property might be infinitely large, so my original claim that they are similar isn't true.

---

> > > ### Author Response · Authors · 2021-11-17
> > > **Discussion of part 2 of response**
> > >
> > > Once again we thank the reviewer for their
> > > valuable comments.
> > > We answer the questions raised in their review.
> > >
> > > >Isn't it true that the rule that is shown to exist is "designed" to exactly match the subgraph of D
> > > that the GNN observed? It's perhaps an oversimplification on my part to say that the rule "represents" the subgraph, but as far as I understood it, the construction seems to say that the rule only applies to subgraphs (of perhaps some other dataset $D'$) that are isomorphic to one we found in D.
> > >
> > > It is not true that the rule will only apply to datasets $D'$ that are isomorphic to a subgraph of $D$.
> > > Please note that a rule $r$ applies to a dataset $D$ if
> > > the body of $r$ can be _homomorphically embedded_ to $D$. This includes the case when $D$ contains a sub-dataset (i.e., a subgraph) _isomorphic_ to the body of $r$, but also many other cases.
> > > As a result, $r$ may apply to datasets $D$ and $D'$ where their relevant subsets witnessing the application of $r$ are neither isomorphic nor homomorphically embedding to each other.
> > >
> > > For example, using the procedure in the proof of Theorem 10 we may extract
> > > a rule $A(x) \gets R(x,z_1) \wedge S(x,z_2)$ for a
> > > fact $A(a) \in T_{\mathcal{M}}(D)$ with $D =\\{ R(a,a), S(a,b)\\}$.
> > > On the one hand, the rule applies to $D$ with the substitution $\\{x \mapsto a$, $z_1 \mapsto a$, $z_2 \mapsto b\\}.$
> > > On the other hand, the rule also applies to a dataset
> > > $D' = \\{ R(a,b), S(a,a)\\}$ (with substitution $\\{x \mapsto a$, $z_1 \mapsto b$, $z_2 \mapsto a\\}$) even though
> > > $D$ and $D'$ are not isomorphic and there is also no homomorphism from $D$ to $D'$ or vice versa.
> > >
> > > >This may just be a notation issue. I understand that rules are not identical to particular subsets of a specific dataset, but I'm using "subgraph" loosely to refer to a configuration of labeled nodes and edges. So when I said "collect every subgraph" I meant "enumerate every configuration of labeled nodes and edges up to the receptive field of the GNN". Perhaps a more precise way to express what I mean would be to say "for every possible subgraph (up to a fixed size) of every possible dataset, up to isomorphism between subgraphs, construct a rule that applies if and only if the dataset graph contains a subgraph isomorphic to this one". Although thinking about it again, I guess perhaps "isomorphic" is too strong, and it would be better to say "homomorphic" here.
> > >
> > > Please, refer to our response to your previous comment. As shown in the example in our
> > > previous comment, a rule can apply to datasets $D$ and $D'$ where their relevant subsets
> > > witnessing the application of the rule are neither isomorphic nor homomorphic to each other.

---

> ### Author Response · Authors · 2021-11-16
> **Response to reviewer Eofg - part 3 of 4**
>
> _(this is a continuation of our response to the review)_
>
> &nbsp;
> >I am not well versed in the knowledge base completion literature, but as far as I know the idea of constraining a GNN to be monotonic under homomorphisms of knowledge bases, and thus to behave like Datalog rules, is novel. However, there has been a fair amount of related work on neural networks that are monotonic for other purposes (and which use similar techniques of nonnegative matrices and monotonic activation functions, among others). As a few examples: [cited works]
>
> >The technical details of the rule extraction mechanism seem novel as well, but as I've stated above, I'm not particularly convinced that the rules it extracts are particularly meaningful, so it's difficult to tell if this is significant.
>
>
> We thank the reviewer for the references provided. Please note, however, that
> these works concern the learning of functions that are monotonic with respect to the standard ordering of the real numbers in Mathematics.
> Our work, however, concerns the learning of functions that are monotonic with respect to the ordering corresponding to the existence of a homomorphism between datasets, which is the standard notion of monotonicity in Logic and theory of Data Management.
> These two notions of monotonicity are rather different. We do appreciate, however, that these works also propose the use
> of  non-negative matrices and monotonic activation functions, and we add a brief discussion about them
> in the revised version of the paper.
>
>  &nbsp;
> >The authors evaluate their approach compared to two existing knowledge graph completion baselines. From the results, it seems like their method is not strictly better or worse than either of the baselines they present, and the different methods have a wide range of performance across the different datasets. One thing that would be useful would be to report standard deviation measures over different random seeds and training splits (specifically the 9:1 split used to construct the incomplete knowledge base and completion targets).
>
>
> This information is added in the revised version of the paper.
>
>  &nbsp;
> >The authors note that the expressivity of MGNNs is limited by the power of "(c,d)-tree-like" rules, and that consequently they cannot express properties such as transitivity. They point to this as a possible reason for the low recall of their method on many of the datasets. They also include an experiment with an alternative test set where they ensure every output fact is at least structurally possible to be classified using their method, and show that recall improves; however, they don't report baseline performance for this test set so it's possible some of the improvement is just because this new test set is easier.
>
> Please note that reviewer PNwA also requested these results, and
> that we have provided a detailed response  to their comment.
> As we point out in our response to them, MGNNs *considerably outperform* the baselines on the alternative splitting, as this way of splitting the data
> favours our system; however, we believe that having this
> comparison in Section 4 would be unfair towards the baselines, and this is why we didn't report them.
> We would be happy to provide these results here or in an Appendix, if either reviewer would like to see them.
>
> &nbsp;
> >The authors also mention that they analyzed some of the rules by hand. I think it would be useful to include additional details of this, for instance by showing some of the extracted rules. I also wonder how close their extracted rules come to capturing the behavior of the MGNN; how many classifications of the MGNN can be explained using the extracted rules, and how many of them are left unexplained due to the truncation on the search method?
>
> In Section 4, subsection ''Extracting Datalog Rules'', we provide some examples of extracted rules.
> For more examples, please note that the supplemental material includes a list of all rules captured in benchmark WN18RRv1 in _rules/extracted/GraIL-BM\_WN18RR\_v1\_from-data\_EC\_0\_extrarules.txt_. It also includes detailed instructions
> in _README.md_ for extracting rules from trained models for all other benchmarks.
>
> Please note also that if a prediction is not explained by the set of rules with one or two body atoms that we
> extract in Section 4,
> a user can easily apply the procedure of Theorem 10 to find a rule that explains it. Therefore, a user can always
> find an explanation for any prediction of interest.
>
>
> If time permits, we will compare the list of predictions made by the truncated set of extracted rules with the list of predictions by the MGNN, and add the results to the revised version of the paper.

---

> > ### Comment · Reviewer_Eofg · 2021-11-17
> > **Discussion of part 3 of response**
> >
> > > These two notions of monotonicity are rather different. We do appreciate, however, that these works also propose the use of non-negative matrices and monotonic activation functions, and we add a brief discussion about them in the revised version of the paper.
> >
> > I agree the concepts are not identical (although they are quite similar), and that the applications are different. My point was just that monotonicity in the logic sense is here enforced using monotonicity (of the neural net outputs) in the mathematical sense, and thus that other works that use monotonicity in the mathematical sense are relevant. Thanks for adding the discussion.
> >
> > > MGNNs considerably outperform the baselines on the alternative splitting, as this way of splitting the data favours our system; however, we believe that having this comparison in Section 4 would be unfair towards the baselines, and this is why we didn't report them. We would be happy to provide these results here or in an Appendix, if either reviewer would like to see them.
> >
> > This is useful to know, thanks. I'd be happy either with adding the results to the appendix (but briefly mentioning in the paper that they are not representative), or leaving them out but adding a textual discussion similar to the one here or to the one in the response to reviewer PNwA. In either case it would be good for readers to know that baselines do not do as well on this split, regardless of whether the numbers are included or not.
> >
> >
> > > In Section 4, subsection ''Extracting Datalog Rules'', we provide some examples of extracted rules.
> >
> > Ah, so the rules described in the paragraph starting with "By analysing the rules, we noted that our transformation learned many rules describing sound inferences" are rules extracted from the MGNN? Sorry for the confusion, I think I misinterpreted those as just examples of what a "symmetric" rule is (for instance), not as an actual extracted rule from the system.
> >
> >
> > > Please note also that if a prediction is not explained by the set of rules with one or two body atoms that we extract in Section 4, a user can easily apply the procedure of Theorem 10 to find a rule that explains it. Therefore, a user can always find an explanation for any prediction of interest.
> >
> > Wouldn't Theorem 10 construct a rule that basically says "if any dataset contains every possibly-relevant fact that this particular dataset contains, then it will be classified as true"? That doesn't seem like a good explanation for the particular dataset we asked about, because the constructed rule may refer to irrelevant facts that didn't actually change the prediction.
> >
> > As an example, if both $A(c)$ and $B(c)$ are in the (test-time) dataset $D$, and the MGNN classifies some other fact $C(c)$ as true, won't the explanation generated by Theorem 10 refer to both $A(c)$ and $B(c)$, even if one of those could be removed without changing the output of the MGNN?

---

> > > ### Author Response · Authors · 2021-11-17
> > > **Discussion of part 3 of response**
> > >
> > > >This is useful to know, thanks. I'd be happy either with adding the results to the appendix (but briefly mentioning in the paper that they are not representative), or leaving them out but adding a textual discussion similar to the one here or to the one in the response to reviewer PNwA. In either case it would be good for readers to know that baselines do not do as well on this split, regardless of whether the numbers are included or not.
> > >
> > > We thank the reviewer for pointing this out. We will make sure that this is discussed in the final version of the paper.
> > >
> > > >Ah, so the rules described in the paragraph starting with "By analysing the rules, we noted that our transformation learned many rules describing sound inferences" are rules extracted from the MGNN? Sorry for the confusion, I think I misinterpreted those as just examples of what a "symmetric" rule is (for instance), not as an actual extracted rule from the system.
> > >
> > > Yes, these are examples of rules that our system extracted from the models trained for the benchmarks in Section 4.
> > >
> > > >Wouldn't Theorem 10 construct a rule that basically says "if any dataset contains every possibly-relevant fact that this particular dataset contains, then it will be classified as true"? That doesn't seem like a good explanation for the particular dataset we asked about, because the constructed rule may refer to irrelevant facts that didn't actually change the prediction.
> > > >As an example, if both $A(c)$ and $B(c)$ are in the (test-time) dataset $D$, and the MGNN classifies some other fact $C(c)$
> > > as true, won't the explanation generated by Theorem 10 refer to both $A(c)$
> > > and $B(c)$, even if one of those could be removed without changing the output of the MGNN?
> > >
> > > Suppose that both $A(c)$ and $B(c)$ are in the (test-time) dataset $D$,
> > > and MGNN $\mathcal{M}$ predicts (i.e., classifies as true) a fact $C(c)$.
> > > Suppose a user is interested in understanding why $C(c)$ is predicted.
> > > The proof of Theorem 10 shows how to construct a rule $r$
> > > with head $C(x)$ such that
> > > $r \in \mathcal{P}\_{\mathcal{M}}$
> > >  and $C(c) \in T_r(D)$.
> > > The reviewer is right in that this rule will contain atoms $A(x)$ and $B(x)$
> > > in the body.
> > > However, a user can remove $A(x)$ or $B(x)$ (or both) from the body of $r$,
> > > and check whether the resulting rule is still captured by
> > >  $T_{\mathcal{M}}$ (using Proposition 7), and whether it still derives $C(c)$ from $D$.
> > >  If so, the user will have obtained a shorter explanation of why $C(c)$ is derived from $D$.

---

> > > > ### Comment · Reviewer_Eofg · 2021-11-17
> > > > **Discussion of part 3 of response**
> > > >
> > > > > However, a user can remove $A(x)$ or $B(x)$ (or both) from the body of $r$, and check whether the resulting rule is still captured by $T_\mathcal{M}$ (using Proposition 7), and whether it still derives $C(c)$ from $D$. If so, the user will have obtained a shorter explanation of why $C(c)$ is derived from $D$.
> > > >
> > > > This makes sense, and I think it would be useful to add some discussion of this procedure to the revised paper. In my opinion, the original rule extracted by Theorem 10 in this case doesn't count as an "explanation" because it refers to things that aren't actually relevant, but if the rule was then pruned so that it was minimal (in the sense that removing any of the facts mentioned by the rule from the dataset would cause the prediction to change), that seems like it could qualify as an explanation of what caused the prediction.
> > > >
> > > > One question on this: is there always a single minimal rule that explains any given prediction, or might there be many rules, none of which subsume the others? I think a good explanation for why a given atom was predicted to be true should allow a user to characterize which subsets of facts would change the prediction if they were removed. Pruning a rule iteratively by removing individual atoms seems like it would identify one such subset, but I wonder if different orders of pruning would lead to different final explanations, and so a full explanation would have to be a set of such rules? (For instance, maybe there are situations where any two out of three facts are sufficient for a classification; perhaps we would need three rules to characterize this? And that set of three rules would be better than a single rule which required all three facts, since that's not actually reflective of the behavior of the model.)
> > > >
> > > > Perhaps this relates to the subsumption question in the part 4 thread; it would be nice if we could extract exactly the set of rules that apply to a given test time query and which are also not subsumed by any other rules. Is this possible, and have you tried to do this to explain individual predictions of the MGNN?

---

> > > > > ### Author Response · Authors · 2021-11-18
> > > > > **Discussion of part 3 of response**
> > > > >
> > > > > >This makes sense, and I think it would be useful to add some discussion of this procedure to the revised paper.
> > > > >
> > > > > We thank the reviewer for this suggestion. We include a discussion of the procedure in the revised paper.
> > > > >
> > > > > >One question on this: is there always a single minimal rule that explains any given prediction, or might there be many rules, none of which subsume the others?
> > > > >
> > > > > This isn't necessarily such minimal rule. For instance, $T_{\mathcal{M}}$ may capture two rules $A(x) \to C(x)$ and $B(x) \to C(x)$, both of which derive $C(c)$ from a dataset $\\{A(c),B(c)\\}$, but neither of which subsumes the other.
> > > > >
> > > > > >Pruning a rule iteratively by removing individual atoms seems like it would identify one such subset, but I wonder if different orders of pruning would lead to different final explanations, and so a full explanation would have to be a set of such rules?
> > > > >
> > > > > Yes, different orders of pruning may lead to different rules. For example, if a user has proved that the $T_{\mathcal{M}}$ from above captures $A(x) \wedge B(x) \to C(x)$, and then they prune $A(x)$, they will discover that $T_{\mathcal{M}}$ captures  $B(x) \to C(x)$; however, if they prune $B(x)$ instead, they will discover that $T_{\mathcal{M}}$ captures  $A(x) \to C(x)$.
> > > > >
> > > > > >Perhaps this relates to the subsumption question in the part 4 thread; it would be nice if we could extract exactly the set of rules that apply to a given test time query and which are also not subsumed by any other rules. Is this possible, and have you tried to do this to explain individual predictions of the MGNN?
> > > > >
> > > > > Our implementation extracts a program $\mathcal{P}$ that subsumes $\mathcal{P}\_{\mathcal{M}}$, with the property that no rule in $\mathcal{P}$ is subsumed by another rule in $\mathcal{P}$. We have not implemented, however, the procedure in the proof of Theorem 10 which identifies a rule captured by $T_{\mathcal{M}}$ deriving a given fact from a given dataset. Developing an optimised implementation of that procedure would be a very interesting task for future work.

---

> > ### Author Response · Authors · 2021-11-18
> > **Discussion of part 3 of response**
> >
> > Further to our previous comment, we have performed several runs with different training splits,
> > and observed that the standard deviations in the metrics are small, in the order of 0.1\%.
> > For a final version of the paper, with additional time, we are happy to extend the evaluation and repeat all experiments in similar conditions over a larger number of training splits, and report average metrics and standard deviations.

---

> ### Author Response · Authors · 2021-11-16
> **Response to reviewer Eofg - part 4 of 4**
>
> _(this is a continuation of our response to the review)_
>
> &nbsp;
> >I also notice that their method learns orders of magnitude more rules than AnyBURL, e.g. their method learns roughly 90,000 rules on FB15K-237 v4, whereas AnyBURL achieves higher accuracy with only 3,574 rules. It seems like fewer rules should be better for a good explanation, right?
>
>
> The purpose of our method is to extract a set of rules that completely describes the behaviour of a neural network. In contrast, AnyBURL
> attempts to produce a minimal set of rules with maximal accuracy for KG completion (note that AnyBURL is a
> a data-dependent rule extraction system, which does not involve training an ML model). These are two very different goals, and therefore we feel that
> contrasting
> the numbers of rules in Table 4 with those in Table 5 is not a meaningful comparison.
>
> &nbsp;
> >Here the authors seem to be working under the assumption that having the classification come from a collection of Datalog rules makes it a "better" choice than using an ordinary GNN for a credit application decision. But as I discuss in my main review, I don't think their method necessarily extracts a good explanation for the behavior of the system.
>
>
> We do not work under such an assumption. The advantage of our approach with respect to a standard GNN is that for every prediction of the network, a user can extract a Datalog rule that is provably captured by the GNN and suffices to generate the prediction.
> As we discuss in response to a previous comment, this property cannot be achieved for general GNNs.
>
>
> Being able to extract rules that are *mathematically proven* to be responsible for predictions is a way to verify *formally* whether the
> decision algorithm of a GNN conforms to any given regulations or standards, as we explain in paragraph 3 of Section 1.
>
> &nbsp;
> >Furthermore, I don't think that being able to explain the conclusions in this manner is sufficient for addressing the complex societal issues for a decision such as credit application. I'm not sure that this warrants a full ethics review, but I am flagging it because it stood out to me (and because the authors do not include any ethics discussion in the paper). I think it would be good for the authors to expand on the ethical implications of their approach if applied to situations like this, and in particular on how an explanation in terms of Datalog rules might be used in practice. (For instance, is the idea that a human would inspect the generated rules for a particular graph to resolve a credit application dispute? Or inspect them in advance to determine whether the model is doing something unfair? Are either of these actually feasible for the rulesets generated by the proposed method, given the extremely large number of rules?
>
>
> We do not claim in our paper that our approach addresses all complex societal issues for a decision such as credit application.
> In fact, we do not believe that any purely algorithmic solution will ever be able to address these complex issues on its own.
> The application of an MGNN to a credit application and subsequent extraction of rules can be used an additional source
> of information that can *support* decision making. We have rephrased the relevant sentence in the revised version of the paper to
> clarify this.

---

> > ### Comment · Reviewer_Eofg · 2021-11-17
> > **Discussion of part 4 of response**
> >
> > > These are two very different goals, and therefore we feel that contrasting the numbers of rules in Table 4 with those in Table 5 is not a meaningful comparison.
> >
> > I see, so AnyBURL is explicitly designed to generate a minimal set of rules, whereas your system is not designed to do that? I suppose this makes sense. Not necessary to change, but it might be nice to call this out explicitly, since otherwise Table 5 seems like a natural point of comparison for Table 4.
> >
> > > Being able to extract rules that are mathematically proven to be responsible for predictions is a way to verify formally whether the decision algorithm of a GNN conforms to any given regulations or standards, as we explain in paragraph 3 of Section 1.
> >
> > I would appreciate if the authors could expand on this point. It's not obvious to me that a set of (thousands of automatically-extracted) Datalog rules would allow formally verifying whether predictions conform to regulations or standards (but perhaps there is a straightforward way to do this that I'm not aware of).
> >
> > For instance, suppose you wanted to show that your classification system never makes a decision based on a protected attribute such as religion. Even if the MGNN never makes a decision based on religion (in the sense that it makes the same decision no matter which religion a person has), the MGNN still "captures" rules that refer to religion, and in particular for every rule $B(x) \leftarrow A(x)$ that it captures, it also captures $B(x) \leftarrow A(x) \land hasReligionFoo(x)$. So it would be entirely possible for your system to produce $B(c) \leftarrow A(c) \land hasReligionFoo(c)$ as an explanation for the prediction $B(c)$ in a specific dataset where both $A(c)$ and $hasReligionFoo(c)$ are known.
> >
> > It also seems that it might be computationally infeasible to process the rules in this kind of a setting. In order to verify the behavior formally, wouldn't you need to extract every rule (e.g. not just those with at most two body atoms), to ensure that none of the rules are problematic?

---

> > > ### Author Response · Authors · 2021-11-17
> > > **Discussion of part 4 of response**
> > >
> > > >It's not obvious to me that a set of (thousands of automatically-extracted) Datalog rules would allow formally verifying whether predictions conform to regulations or standards (but perhaps there is a straightforward way to do this that I'm not aware of).
> > > For instance, suppose you wanted to show that your classification system never makes a decision based on a protected attribute such as religion. Even if the MGNN never makes a decision based on religion (in the sense that it makes the same decision no matter which religion a person has), the MGNN still "captures" rules that refer to religion, and in particular for every rule $B(x) \gets A(x)$ that it captures, it also captures
> > > $B(x) \gets A(x) \wedge hasReligionFoo(x)$. So it would be entirely possible for your system to produce
> > > $B(c) \gets A(c) \wedge hasReligionFoo(c)$ as an explanation for the prediction
> > > $B(c)$ in a specific dataset where both $A(c)$ and $hasReligionFoo(c)$ are known.
> > > It also seems that it might be computationally infeasible to process the rules in this kind of a setting. In order to verify the behavior formally, wouldn't you need to extract every rule (e.g. not just those with at most two body atoms), to ensure that none of the rules are problematic?
> > >
> > > The reviewer is right in pointing out that, if rule $B(x) \gets A(x)$ is captured, then
> > > rule $B(x) \gets A(x) \wedge hasReligionFoo(x)$ is also captured.
> > > Note, however, that the latter rule is subsumed (logically implied) by the former.
> > > Although both
> > > rules are captured, the rule extraction algorithm that we use in Section 4
> > > would only generate the first rule,
> > > since rules subsumed by other rules are not included in its output.
> > > Please note that in the revised version of the paper,
> > > we describe explicitly this rule extraction algorithm.

---

> ### Comment · Reviewer_Eofg · 2021-11-19
> **Raised score**
>
> I have updated my review and raised my score from 5 to 6. Thank you for answering my questions and describing the approach and experiments in more detail, although I still think the paper would benefit from a more thorough discussion in the main paper of how to explain predictions in practice (see my updated review).

---

> > ### Comment · Reviewer_Eofg · 2021-11-22
> > **Raising score again**
> >
> > I have raised my score from 6 to 8; thank you for your additional clarifications and for the new experiment regarding the extracted rules. I have a few more suggestions that I think would be important to add in the next revision of the paper (reproduced from my updated review):
> >
> > > - I would recommend briefly mentioning the computational constraints on the current rule-extraction algorithm in the introduction, to make it clear that there is a difference between the theoretical and practical ability to extract rules. Specifically, after the sentence "we can explicate the rules that are implicitly captured by the model and use them to fully explain each credit recommendation", I'd suggest adding another sentence stating that the existing algorithm for extracting rules may only explain a subset of predictions of the original MGNN (and, optionally, that the extracted rules could just be used directly if it is critical to be able to explain every prediction).
> > > - In section 4 "Rule Extraction" after "we applied $T_P$ to the test dataset of each benchmark", it would be useful to explicitly refer to Table 6 in the appendix, so that readers know where to find that information.
> > > - I think it would be important to include some results for AnyBURL and DRUM in Table 6 as well, by extracting a set of rules from them and then evaluating those rules. I am aware neither AnyBURL or DRUM has any theoretical guarantees about extracted rules, but as far as I understand it each method does produce some rules with thresholds, so it seems fair to take a subset of their rules (with threshold similarly tuned for accuracy). This would help answer the question of how much the MGNN adds over the baselines if you require that every prediction be explainable by rules.

---

> > > ### Author Response · Authors · 2021-11-23
> > > **New version of paper uploaded**
> > >
> > > We thank the reviewer for their valuable suggestions. We have uploaded a new revision of the paper that includes the changes suggested in the first and second bullet points. We will address the third suggestion in the camera-ready version, as it requires running new experiments, and the deadline for updating the paper draft is imminent.

---

> ### Author Response · Authors · 2021-11-19
> **Response to revised review.**
>
> We thank the reviewer for their updated review and additional comments.
> We present below our response.
>
> &nbsp;
> >I still feel that the paper could be improved by adding more of a discussion in the main paper of how to actually make use of the extracted rules to explain predictions made by a MGNN in practice. For instance, given the extracted set of rules, how do we formally verify properties of interest about the MGNN? Or, given a particular prediction of the MGNN in a particular dataset, how can we identify which facts the MGNN actually used to make its prediction? Can the full set of rules (e.g. those with more than two body atoms) be extracted quickly enough to make either of these practical?
> These details are especially important because the paper motivates the approach based on the trustworthiness/compliance/verifiability/fairness of the system.After discussion with the authors I believe doing this kind of verification and inspection should be possible in principle (although I have some concerns about runtime), but many of the details are buried in the appendix, and I think they should be given more of a focus.
>
> State-of-the-art Datalog reasoning engines
> provide powerful methods for generating human-readable proofs.
> Specifically, given a set of rules, a dataset, and a fact that
> follows logically from the rules and the data,
> Datalog engines can compute a
> (shortest) proof for that fact, thus explaining why the fact follows logically
> from the rules and the data.
> Such step-by-step proofs can be very useful
> human-friendly explanations.
> Thus, once we have extracted a Datalog program $\mathcal{P}\_{\mathcal{M}}$
> which derives the same facts as $T_{\mathcal{M}}$, for $\mathcal{M}$ an MGNN, we can
> use a Datalog engine to explain predictions,
> and we can even guarantee that the produced explanations are the shortest possible.
> For an example of the kinds of proofs that can be provided by existing systems,
> see [Section 6.8](https://docs.oxfordsemantic.tech/reasoning.html#explaining-reasoning-results)
>  in the documentation
> of RDFox, a state-of-the-art reasoning engine supporting Datalog.
> If the reviewer feels like pointing this out explicitly would be useful,
> we are happy to mention this in the introduction.
>
> &nbsp;
> >However, I'm not convinced that the rules they extract really count as "explanations": as far as I can tell the extracted rule sets can be extremely complex.
>
> Please note that the rules extracted from an MGNN model describe _precisely_ the transformation realised by it.
> If the extracted rules are complex,
> this means that the transformation realised by the MGNN is complex.
> In this case, it is an advantage of our system, rather than a disadvantage, that we are able to fully
> capture all complex details of this transformation.
> No other system in the literature that we are aware of is capable of doing this.
>
> &nbsp;
> >Empirical results are also not particularly strong; their system achieves comparable performance to other methods but seems to require orders of magnitude more rules [...]
>
> Please note that as pointed out in a previous comment to the reviewer, we feel that contrasting the number of rules extracted by different systems is not a meaningful comparison because the purpose of extracting rules in each system is very different. Please note also that in Section 4, we make the predictions for the benchmarks using directly
> the MGNN models, so the number of rules captured by the models is irrelevant to the performance metrics.
>
> &nbsp;
> > [...] and also has fundamental limits to its expressivity.
>
> Please note that this is also a limitation of the baselines. As we point out in Section 4, baselines learn exclusively
> chain rules. This means, for example, that they cannot learn inverse rules like $S(x,y) \gets S(y,x)$, which can be learned by our system.
>
> &nbsp;
> >Update: After discussion with the authors, I have raised my score from 5 to 6; some of my concerns in my initial review were based on a misunderstanding of the proposed method, and the authors have addressed many of my comments regarding clarity and experimental details in the revised paper. I still feel that the paper could be improved by describing in more detail how to use the extracted rules to explain predictions, since this seems to be the main way their approach differs from prior work.
>
> We feel that the main way our approach differs from prior work is that we can extract
> a set of rules that _provably_ and _exactly_ characterises the transformation
> between datasets induced by an ML model. This result is the main
> subject of Section 3, where we discuss it thoroughly.
>
> Please note that once such set of rules has been extracted, one can easily use
> symbolic rule engines to produce explanatory proofs, as we point out in a previous comment
> in this response.

---

> > ### Comment · Reviewer_Eofg · 2021-11-19
> > **Discussion**
> >
> > > For an example of the kinds of proofs that can be provided by existing systems, see Section 6.8 in the documentation of RDFox, a state-of-the-art reasoning engine supporting Datalog. If the reviewer feels like pointing this out explicitly would be useful, we are happy to mention this in the introduction.
> >
> > Thank you for the reference. I do think pointing this out would be useful, as I was not very familiar with the capabilities of Datalog reasoning engines.
> >
> > This only addresses part of my concern, though, because the optimized extraction algorithm described in the paper still only extracts rules with up to $m$ body atoms. If I understand correctly, a system like RDFox could explain a prediction using the subset of extracted rules, but that might not be fully representative of the MGNN's predictions. The MGNN might make a true prediction that isn't explained by any of the rules, for instance; or, if a user is interested in all of the reasons why the MGNN classified as true (not just the shortest proof), RDFox might miss some due to them being outside of the set. In the first case, one could apply Theorem 10 to obtain one proof, but that proof is probably not the shortest one, because the rule may be subsumed by other rules we haven't extracted. (The modification described in the revised appendix for searching for shorter explanations from Theorem 10 seems like it could be part of something like this, but it's not described in much detail and I'm not sure it's enough to extract every possible proof.)
> >
> > A similar issue arises when trying to formally verify behavior of the MGNN, because it seems that verifying behavior might involve identifying situations where the MGNN provably does NOT make a prediction, or proving that no situation exists in which a "unfair" prediction is made. I don't think you can do this without extracting the full set of rules captured by the MGNN, right? But the extraction algorithm in the paper only extracts a subset. (Moreover, can Datalog reasoning engines answer these kinds of questions? For instance, can you answer the question "is there any set of facts for which adding a fact like $hasReligionX(c)$ causes a prediction to change from False to True?")
> >
> > If it is computationally infeasible to extract all of the rules, it seems that relying on a Datalog reasoning engine is insufficient, because the Datalog reasoning engine is not operating on the full set of behaviors of the MGNN. Perhaps there are other algorithms you can use to verify particular properties without extracting all of the rules (e.g. based on Theorem 10), but if so, I think those algorithms should be explicitly discussed in the paper.
> >
> > (Actually, this discussion raises an interesting question. How many of the predictions made by the MGNN are explained by the extracted rules with only two body atoms? If the extracted rules alone have good performance, that might indicate that considering rules with two body atoms is a good approximation of the transformation between datasets learned by the MGNN. One could even use the learned rules directly, if they were worried about rare behavior of the MGNN that is not captured by the extracted rules. It would also be interesting to compare this to a similar set of best-effort rules extracted for DRUM or AnyBURL; perhaps MGNN would suffer less from "discretizing" into a set of extracted rules due to its theoretical properties.)

---

> > > ### Author Response · Authors · 2021-11-20
> > > **Discussion of revised review**
> > >
> > > >Thank you for the reference. I do think pointing this out would be useful, as I was not very familiar with the capabilities of Datalog reasoning engines.
> > >
> > > We thank the reviewer for the suggestion. We will shortly upload a new version of the paper with this modification.
> > >
> > > &nbsp;
> > > >This only addresses part of my concern, though, because the optimized extraction algorithm described in the paper still only extracts rules with up to $m$ body atoms.
> > >
> > >
> > > Please note that according to Definition 9, for any MGNN $\mathcal{M}$, there exists an upper bound to the number of atoms in the body of a rule in program $\mathcal{P}\_\mathcal{M}$. Therefore, by setting $m$ to
> > > such bound, we can ensure that the algorithm extracts a program deriving the same facts as
> > > $T_{\mathcal{M}}$ on any dataset.
> > >
> > > &nbsp;
> > > >if a user is interested in all of the reasons why the MGNN classified as true (not just the shortest proof), RDFox might miss some due to them being outside of the set.
> > >
> > > Please note that given a Datalog program, a dataset, and a fact derived by the program from the dataset, RDFox
> > > can be configured to also
> > > produce _all_ explanatory proofs of the fact from the program and the dataset, not just a shortest one.
> > >
> > > &nbsp;
> > > >A similar issue arises when trying to formally verify behavior of the MGNN, because it seems that verifying behavior might involve identifying situations where the MGNN provably does NOT make a prediction, or proving that no situation exists in which a "unfair" prediction is made. I don't think you can do this without extracting the full set of rules captured by the MGNN, right? But the extraction algorithm in the paper only extracts a subset.
> > >
> > > Please note that, as we point out in a response to a previous comment, the optimised extraction algorithm in the paper can extract
> > > a set of rules that fully characterises the transformation between datasets induced by an MGNN.
> > >
> > > &nbsp;
> > > >(Moreover, can Datalog reasoning engines answer these kinds of questions? For instance, can you answer the question "is there any set of facts for which adding a fact like $hasReligionFoo(c)$ causes a prediction to change from False to True?")
> > >
> > > Yes, this question can be simply answered by inspecting the rules. If the body of a rule is of the form $C \wedge hasReligionFoo(x)$, for $C$ _any_ conjunction, then any grounding of $C$ is a dataset such that adding a fact like $hasReligionFoo(c)$ changes a prediction from False to True.
> > > Conversely, if $hasReligionFoo(x)$ does not appear in the body of any rule, then clearly adding a fact like $hasReligionFoo(c)$ cannot change the output of the program on any dataset.
> > >
> > > &nbsp;
> > > >How many of the predictions made by the MGNN are explained by the extracted rules with only two body atoms? If the extracted rules alone have good performance, that might indicate that considering rules with two body atoms is a good approximation of the transformation between datasets learned by the MGNN. One could even use the learned rules directly, if they were worried about rare behavior of the MGNN that is not captured by the extracted rules. It would also be interesting to compare this to a similar set of best-effort rules extracted for DRUM or AnyBURL; perhaps MGNN would suffer less from "discretizing" into a set of extracted rules due to its theoretical properties.
> > >
> > > We thank the reviewer for this suggestion. We have used the extracted programs (where all rules have at most two body atoms) to classify the positive and negative examples in the test dataset of each benchmark. We have observed that precision metrics remain the same, while
> > > recall decreases only very slightly (1.6\% in average). This suggests that the extracted rules suffice to account for almost all
> > > predictions made by the MGNN models that were learned on the benchmarks, which is a very interesting result. In Section 4 of the new version of the paper, we have included a description of this experiment and the results that we obtained. We also present all classification metrics obtained in the experiment in a new table in Appendix D.

---

> > > > ### Comment · Reviewer_Eofg · 2021-11-20
> > > > **Clarifications**
> > > >
> > > > > > if a user is interested in all of the reasons why the MGNN classified as true (not just the shortest proof), RDFox might miss some due to them being outside of the set.
> > > > >
> > > > > Please note that given a Datalog program, a dataset, and a fact derived by the program from the dataset, RDFox can be configured to also produce all explanatory proofs of the fact from the program and the dataset, not just a shortest one.
> > > >
> > > > My point was that RDFox can only do this if it is given the full set of rules that characterize the MGNN's behavior; it would not be able to produce proofs involving rules with three body atoms, for instance, if given only the subset of rules with two body atoms.
> > > >
> > > > > Therefore, by setting $m$ to such bound, we can ensure that the algorithm extracts a program deriving the same facts as $T_\mathcal{M}$ on any dataset.
> > > >
> > > > I understand that this is theoretically possible. My question was whether it was computationally feasible, since that determines whether the MGNN's behavior could actually be fully verified in practice. Have you actually done this?

---

> > > > > ### Author Response · Authors · 2021-11-22
> > > > > **Discussion of revised review**
> > > > >
> > > > > >I understand that this is theoretically possible. My question was whether it was computationally feasible, since that determines whether the MGNN's behavior could actually be fully verified in practice. Have you actually done this?
> > > > >
> > > > > We thank the reviewer for the clarification. Currently, for the benchmarks in Section 4, we need to limit the number of atoms in bodies of rules that we consider in the extraction algorithm. We believe, however, that there are many ways in which the rule extraction algorithm can be optimised, and we will explore them in future research. Please note also that, as our recent experiment shows, the rules that we can extract already account for almost all predictions made by the MGNN models on the benchmarks.

---

### Official Review · Reviewer_PNwA · 2021-11-01

**Correctness:** 3
**Technical Novelty And Significance:** 2
**Empirical Novelty And Significance:** 2
**Recommendation:** 5
**Confidence:** 4

**Main Review:**

Pros:

* It first transforms the original knowledge graph into a colored graph where each vertex is labeled by a feature vector and represents the facts of the corresponding entity or entity pairs. It further connects the vertices that refer to the common entities. Such a transformation is a bit inspiring, and it is naturally suitable for inductive settings.

* The authors make it clear that there is an equivalent set of rules for any MGNN, and that both can produce the same facts. Hence, we can use rules to understand the predictions made by MGNNs.

Cons:

* The approach appears to be straightforward. I discovered that Equation 1 is sufficient to describe the model. Furthermore, Section 3, which is two pages long, only states that there is a set of rules that can be used to derive the same facts as MGNNs. The authors should simplify this section and add some details to the appendix, in my opinion.

* It appears that rule extraction is a searching procedure that requires a lot of space and time. This could limit its use in larger datasets. Some baselines, such as DRUM and AnyBURL, can also capture rules. Although the exact relationship between the predictions and the results of applying the rules to these methods is unknown, the generated rules can still assist us in understanding their predictions to some extent. Table 2 also shows that these baselines can produce competitive results. Hence, I think the advantage of the proposed approach is not obvious.

Questions and Suggestions:

* The authors should add some citations regarding the transformation process into a colored graph, as there exists some work that adopts similar transformations.

* Theorem 10 describes the expressiveness limitation of MGNNs, i.e., MGNNs can only capture some rules. I want to know whether some baselines (e.g., DRUM and AnyBURL) also have this limitation, as they can also capture rules.

* The authors split the testing dataset and re-evaluated their approach in Table 3. I think it is better to provide the results of baselines in this setting.

* In Section 3, the authors said that "we can enumerate all such rules, apply Proposition 7 to each rule, and keep only the rules that pass this check". It seems that the rule extraction procedure is time-consuming. Can the authors provide the corresponding time cost? I also notice that the last column of Table 2 records the training time. I want to know whether the training time includes the time cost of rule extraction.

* In Section 4, the authors said that "to reduce the search space, we considered only rules with at most two body atoms". I think the extracted rules can not exactly describe the predictions made by MGNNs, as the authors only consider a part of the rules.

* From Table 4 and 5, I notice that $T_M$ captures much more rules than AnyBURL on FB15K-237, and fewer rules on WN18RR. Can the authors explain the phenomenon? Moreover, the fourth column in Table 5 shows the number of AnyBURL rules captured by $T_M$ on $S_I$. Can the authors provide the number of rules captured by AnyBURL on $S_I$? It will help readers see the proportions of rules generated by AnyBURL and $T_M$.


**Summary Of The Paper:**

This paper proposes a new family of GNN-based transformations for knowledge graphs, which is called monotonic GNNs (MGNNs). The key characteristic of MGNNs is that we can produce an equivalent set of rules for an arbitrary MGNN, and the rules and the MGNN-based transformation can produce the same facts. In other words, the rules can be used to explain how MGNNs make predictions. Moreover, the authors provide dedicated proofs for their formal statements. The experimental results on classification tasks for knowledge graph completion demonstrate the effectiveness of the approach.

**Summary Of The Review:**

The idea of the paper is interesting but the technical contributions are not very significant and the advantage of the proposed approach is not obvious.

---

> ### Author Response · Authors · 2021-11-16
> **Response to reviewer PNwA - part 1 of 2**
>
> We thank the reviewer for their
> valuable comments.
> We next discuss in detail how we have
> addressed them, and
> answer the questions raised in their review.
> We will release a revised version of the paper
> in a few days, which includes all changes discussed below.
>
> &nbsp;
> > The approach appears to be straightforward. I discovered that Equation 1 is sufficient to describe the model.
>
> Our approach consists of three (equally important) elements: the encoder, the MGNN, and the decoder.
> Equation (1) describes neither the encoder nor the decoder, which are both carefully designed and non-trivial. Furthermore,
> it also does not provide a full description of the MGNN, as
> we also need to specify the relevant constraints
> that the matrices, the activation function and the classification function of an MGNN must satisfy (these are given in
> Definition 3).
>
>  &nbsp;
> > Furthermore, Section 3, which is two pages long, only states that there is a set of rules that can be used to derive the same facts as MGNNs. The authors should simplify this section and add some details to the appendix, in my opinion.
>
>  Note that the main result in Section 3 is not only that there exists a set of rules that can be used to derive the same facts as MGNNs, but also a constructive description of these rules (Theorem 10). This constructive description is built in several steps (Definition 5, Definition 6, Proposition 7, Theorem 8, Definition 9), all of which are essential and cannot be removed without making the paper not self-contained.
>
>   &nbsp;
> >It appears that rule extraction is a searching procedure that requires a lot of space and time. This could limit its use in larger datasets.
>
> Please note that the  rule extraction procedure is data-independent; in particular, it
> only depends on the
> _schema_ of the data (that is, the set of relevant unary and binary predicates).
> The size of the datasets we may want to apply the model to is completely irrelevant.
> As mentioned before, the rules characterise the behaviour of the model for _arbitrary datasets_
> and hence are not specific to any paticular input dataset---that is why our main result is rather powerful and non-trivial.
>
>    &nbsp;
> >Some baselines, such as DRUM and AnyBURL, can also capture rules. Although the exact relationship between the predictions and the results of applying the rules to these methods is unknown, the generated rules can still assist us in understanding their predictions to some extent. Table 2 also shows that these baselines can produce competitive results. Hence, I think the advantage of the proposed approach is not obvious.
>
>
> Please note that DRUM's paper does not provide an algorithm for extracting rules from trained models, and DRUM's code does not implement a rule extraction procedure.
> This was already pointed out in DRUM's [GitHub page](https://github.com/alisadeghian/DRUM/issues/2)
> by three different users in 2020, but the authors have not replied; to the best of our knowledge, no rule extraction code for DRUM is currently available.
> Therefore, users cannot verify to what extent the rules learned by DRUM explain
> the predictions of its model. This stands in contrast with our approach, where Theorem 10 establishes an *exact* correspondence *on any dataset* between the predictions of the
> model and the extracted set of rules. This is a crucial advantage of our approach.
> Please also note that, as pointed out in our paper, AnyBURL is a data-dependent rule extraction method which
> does not
> involve training a model; hence, the rules are the only outcome of the procedure
> and there is no corresponding model.
>
>  &nbsp;
> >The authors should add some citations regarding the transformation process into a colored graph, as there exists some work that adopts similar transformations.
>
>
> We are not aware of any work describing an encoding that transforms datasets into coloured graphs in a way similar to our encoding.
> We would be grateful to the reviewer if they could kindly provide references to those works.
>
>  &nbsp;
> >Theorem 10 describes the expressiveness limitation of MGNNs, i.e., MGNNs can only capture some rules. I want to know whether some baselines (e.g., DRUM and AnyBURL) also have this limitation, as they can also capture rules.
>
>
> This is discussed in Section 4, where we explain that DRUM and AnyBURL
> can only generate _chain_ rules, as defined in the paper.

---

> ### Author Response · Authors · 2021-11-16
> **Response to reviewer PNwA - part 2 of 2**
>
> _(this is a continuation of our response to the review)_
>
> &nbsp;
> > The authors split the testing dataset and re-evaluated their approach in Table 3. I think it is better to provide the results of baselines in this setting.
>
> Our experimental results show that MGNNs *significantly outperform* the baselines in almost all benchmarks
> with the alternative splitting used for Table 3.
> However, we felt that presenting these results in the paper would be unfair towards the baselines.
> Indeed, please note that our encoding ensures that all positive examples in the
> alternative splitting can potentially be derived by an MGNN, but we do not know if this also holds for
> the baselines. This gives our system
> an unfair advantage in a comparison.
> We did not wish to misrepresent the effectiveness of our system in our favour.
> Therefore, in Section 4 we only compared our system with the baselines over the
> splitting provided in the literature.  If the reviewer feels that it would be interesting to provide the results of the baselines
> for the alternative splitting that we use for Table 3, we would be happy to provide a table.
>
>   &nbsp;
> >In Section 3, the authors said that "we can enumerate all such rules, apply Proposition 7 to each rule, and keep only the rules that pass this check". It seems that the rule extraction procedure is time-consuming. Can the authors provide the corresponding time cost?
>
>
> Please see our previous comment about the rule extraction procedure.
>
>   &nbsp;
> >I also notice that the last column of Table 2 records the training time. I want to know whether the training time includes the time cost of rule extraction.
>
>
> The training time does not include the time cost of rule extraction, since the process of rule extraction is not part of the training.
> Indeed, rule extraction only applies to models that have already been trained.
> The rule extraction times range from a few seconds (for most benchmarks)
> to about 5 minutes (for the largest FB15K-237 benchmark).
> We are happy to provide a table with the exact times, if the reviewer feels like it would be interesting.
>
>
> &nbsp;
> >From Table 4 and 5, I notice that $T_M$ captures much more rules than AnyBURL on FB15K-237, and fewer rules on WN18RR. Can the authors explain the phenomenon?
>
>
> We believe that the reason for this discrepancy is the selection of different threshold functions $\mathsf{cls}$ during the hyperparameter selection in the validation phase for benchmarks FB15K-237 and WN18RR.
> The thresholds selected for our model on the FB15K-237 family of benchmarks are very low, and therefore the rule extraction algorithm
> generates a high number of rules. In contrast, for the WN18RR benchmarks, the selected thresholds are much higher, and the rule extraction algorithm produces a smaller number of rules.
>
> &nbsp;
> >Moreover, the fourth column in Table 5 shows the number of AnyBURL rules captured by
> $T_M$ on $S_I$. Can the authors provide the number of rules captured by AnyBURL on $S_I$? It will help readers see the proportions of rules generated by AnyBURL and $T_M$.
>
>
> Please note that the number of rules captured by AnyBURL on $S_I$ is provided in the first column of Table 5.

---

> > ### Comment · Reviewer_PNwA · 2021-11-29
> > **Thanks for your response.**
> >
> > Thanks for your response to address most of my concerns. I still think that the proposed method lacks novelty. However, I would not fight for rejection if the paper gets a chance to be accepted.

---

> > > ### Author Response · Authors · 2021-11-30
> > > **Response**
> > >
> > > We thank the reviewer for their response. We would like to emphasise that, to the best of our knowledge, our approach is the first to introduce a class of neural networks learning dataset transformations that: (i) can be entirely characterised as Datalog programs, and (ii) can be trained in practice to achieve competitive performance in tasks of interest such as KG completion. Previous works using neural networks to learn dataset transformations do not (to the best of our knowledge) prove any relation between learned models and logical rules. In fact, as we prove in a response to another reviewer, some transformations learned by (non-monotonic) GNN-based models cannot​ be captured by any first-order program.

---

### Official Review · Reviewer_EWub · 2021-11-02

**Correctness:** 3
**Technical Novelty And Significance:** 2
**Empirical Novelty And Significance:** 2
**Recommendation:** 6
**Confidence:** 4

**Main Review:**

Strengths:

- This paper studies how to improve the interpretability of GNN-based predictions, which is an important research direction.
- The authors provide a detailed theoretical analysis of the model’s ability to
explain the GNN-based predictions by logical rules.

Weaknesses:

In terms of the experiments, my concerns are as follows.
- This proposed method is based on GNN. However, the comparison with GNN-based methods is missing. The authors may want to compare their approach with the state-of-the-art GNN-based ones, such as GraIL[1] .
- MGNN does not outperform some existing models like DRUM and AnyBURL on several benchmarks.
- The authors may want to provide some case studies to show how MGNN ensures that the predictions can be explained by logical rules, as what DRUM [2] does in Section 5.3.

In terms of the proposed model, MGNN assigns a feature vector to each link, which significantly increases the computational cost. This makes the model difficult to apply to large-scale knowledge graphs.

In terms of the writing, my concerns are as follows.
- The necessary visual illustrations of the idea and the model architecture are missing, making this paper hard to follow. The authors may want to provide figures in Sections 2 and 3.
- The authors may want to specify some important details.
    - In Section 2, the authors say that the result of MGNN can be decoded to an output knowledge graph by "inverting the encoder". However, the decoding procedure is unclear.
    - The definitions of $c_3$ and $c_4$ in Paragraph 4 are missing.
    - Some of the statements are confusing. For example, in Section 2, Definition 3 states that $\sigma$ is a mapping from $\mathbb{R}$ to $\mathbb{R}^+$, while the input of $\sigma$ in Eq. (1) belongs to $\mathbb{R}^d, d\geq 1$. Besides, each vertex of the result $M(G)$ is labelled by a boolean number, while in Definition 4 the label is a vector.

[1] Komal K Teru, Etienne Denis, and William L Hamilton. Inductive relation prediction by subgraph reasoning. In ICML, 2020.

[2] Ali Sadeghian, Mohammadreza Armandpour, Patrick Ding, and Daisy Zhe Wang. DRUM: End-To-End Differentiable Rule Mining On Knowledge Graphs. In NeurIPS, 2019.


**Summary Of The Paper:**

The authors propose a Monotonic Graph Neural Network transformation (MGNN) for inductive knowledge graph completion, which allows the predictions to be explained symbolically by logical rules.

**Summary Of The Review:**

This paper proposes a GNN-based method, named MGNN, which allows the predictions to be explained symbolically as logical reference to improve their interpretability. However, MGNN does not outperform existing approaches on some benchmarks, and it may be difficult to apply to large-scale knowledge graphs.

---

> ### Author Response · Authors · 2021-11-15
> **Response to reviewer EWub**
>
> We thank the reviewer for their valuable comments.
> We next discuss in detail how we have addressed them, and answer the questions raised in their review.
>  We will release a revised version of the paper
> in a few days, which includes all changes discussed below.
>
>  &nbsp;
> > In terms of the experiments, my concerns are as follows. This proposed method is based on GNN. However, the comparison with GNN-based methods is missing. The authors may want to compare their approach with the state-of-the-art GNN-based ones, such as GraIL[1].
> MGNN does not outperform some existing models like DRUM and AnyBURL on several benchmarks.
>
>
> The goal of this paper is not to provide the best-performing KG completion model,
> but rather to provide a GNN-based architecture that that is (i)
> potentially applicable to a number of important KG-related tasks (such as KG-completion),
>  (ii) competitive with the state-of-the-art in such tasks and (iii)
>  fully explainable in the sense described in the paper. To validate this, we believe that it suffices to compare
>   with DRUM and Any-BURL as they are state-of-the-art KG completion systems; we also believe that the evaluation
>   results show that our system is competitive with the baselines. An
>   exhaustive comparison with other KG completion methods would, of course, be interesting, but not needed to validate our main claims.
>
> >The authors may want to provide some case studies to show how MGNN ensures that the predictions can be explained by logical rules, as what DRUM [2] does in Section 5.3.
>
>
> In order to validate the claims made in this paper, we do not need to show that rules extracted by the model are judged to be ''reasonable'' by a human observer, since
> this will depend on the particulars of the benchmark. Instead, we show that every prediction made by an MGNN model can be
> produced by a logical rule captured by the model.
> Please note that Theorem 10 in our paper guarantees this mathematically---that is, the rules that we extract capture _precisely_ the predictions of the
> model on _any dataset_.
> In contrast, DRUM provides no such theoretical guarantee.
> Additionally, please note that
> MGNNs are able to extract ''reasonable'' rules, as illustrated by examples in Section 4.
>
>
> >In terms of the proposed model, MGNN assigns a feature vector to each link, which significantly increases the computational cost. This makes the model difficult to apply to large-scale knowledge graphs.
>
>
> Please note that our encoding does not assign a feature vector to each link (binary fact). Instead, each link (binary fact) $S(a,b)$ is represented as a single element of the feature vector for the node $v_{a,b}$ representing the pair of constants $a$ and $b$.
>
>
> >The necessary visual illustrations of the idea and the model architecture are missing, making this paper hard to follow. The authors may want to provide figures in Sections 2 and 3.
>
>
> In the revised version of the paper, we have added figures and new examples in Sections 2 and 3 to make the paper easier to follow.
>
>
> >In Section 2, the authors say that the result of MGNN can be decoded to an output knowledge graph by "inverting the encoder". However, the decoding procedure is unclear.
>
>
> Definition 4 describes the decoding procedure, which is straightforward since there is a one-to-one correspondence between positions in the input and output layer vectors and facts.
> In the revised version of the paper, we have added an example to show how the decoder works.
>
>
> >The definitions of $c_3$ and $c_4$ in Paragraph 4 are missing.
>
>
> Both $c_3$ and $c_4$ are _colours_, as stated at the beginning of the fourth paragraph of Section 2. Furthermore, Definition 2 describes how $c_3$ and $c_4$ are used for encoding of a dataset.
>
>
> >Some of the statements are confusing. For example, in Section 2, Definition 3 states that $\sigma$ is a mapping from $\mathbb{R}$
> to $\mathbb{R}^{+}$, while the input of $\sigma$ in Eq. (1) belongs to $\mathbb{R}^d$, $d \geq 1$.
>
>
> Please note that in the fourth paragraph of Section 2, we state that scalar functions are applied to vectors and matrices element-wise. Therefore, given mapping $\sigma$ and a vector $\mathbf{v} = (v_1, \dots, v_n)$, expression $\sigma(\mathbf{v})$ stands for $(\sigma(v_1),\dots,\sigma(v_n))$.
> In the revised version of the paper, we have clarified this and added a new example.
>
>
> >Besides, each vertex of the result $M(G)$ is labelled by a Boolean number, while in Definition 4 the label is a vector.
>
> Please note that, according to Definition 3, each vertex $v$ in $\mathcal{M}(G)$ is labelled by $\mathsf{cls}(\lambda^L(v))$
> where $\lambda^L$ maps $v$ to a vector of reals and $\mathsf{cls}$ is a function from reals to {$0, 1$}; thus, using our assumption before Definition 1 that we apply scalar functions to vectors and matrices element-wise, we have that
> each vertex in $\mathcal{M}(G)$ is labelled by a {$0, 1$}-vector of dimension $\delta$, which is consistent with Definition 4.

---

> > ### Comment · Reviewer_EWub · 2021-11-25
> > **Discussion**
> >
> > The authors' responses and the updated paper have addressed some of my concerns.
> > However, I have a few more questions to discuss with the authors.
> >
> > (1). The authors only compare the proposed model with DRUM and Any-BURL. However, the datasets used in this paper are originally proposed by GraIL, and DRUM and Any-BURL did not conduct experiments on those datasets. Besides, both GraIL and the proposed MGNN are GNN-based methods for knowledge graph completion tasks. Therefore, the reasons that the authors did not compare MGNN with GraIL are unconvincing.
> >
> >
> > (2). MGNN transforms each pair of entities that co-occurs in a fact from the KG to a vertex in the encoded graph. However, the large number of vertices in the encoded graph may significantly increases the computational cost. The authors may want to analyze the space/time complexities of MGNN and compare with other baseline methods.
> >
> > (3). It is still unclear how the proposed MGNN model predict the relation between two given entities. Take Fig. 1 as an example, how to predict the relation between Crime and Punishment and The Idiot from $G_D$?
> >
> > (4). The authors may want to add some descriptions of the loss function.

---

> > > ### Author Response · Authors · 2021-11-26
> > > **Discussion**
> > >
> > > We thank the reviewer for their comments. Please note that the phase for submitting new revisions of the draft has ended, so any proposed changes will have to be considered in the camera-ready version. We present our responses below.
> > >
> > > &nbsp;
> > > >The authors only compare the proposed model with DRUM and Any-BURL. However, the datasets used in this paper are originally proposed by GraIL, and DRUM and Any-BURL did not conduct experiments on those datasets. Besides, both GraIL and the proposed MGNN are GNN-based methods for knowledge graph completion tasks. Therefore, the reasons that the authors did not compare MGNN with GraIL are unconvincing.
> > >
> > > We have very recently run experiments on GraIL, and we can now report
> > > the results below for GraIL (G) and our MGNN-based system (M), in percentages.
> > > Results on FB15K-237 v3 and v4 and NELL-995 v3 and v4 are not
> > > reported because GraIL takes many days to train on those datasets.
> > >
> > >
> > > Benchmark  | Precision (G) | Precision (M) |  Recall (G) | Recall (M) |  Accuracy (G) | Accuracy (M) |  F1 Score (G) | F1 Score (M) |   AUC (G) | AUC (M) |   Training, in s. (G) | Training, in s. (M) |
> > > ---|---|---|---|---|---|---|---|---|---|---|---|---|
> > > FB15K-237 v1 | 41.5 | 98.5|  92.2|  32.7 | 69.0|  66.1 | 57.1 | 49.1|  78.6 | 65.7 | 23,868 | 9,192|
> > > FB15K-237 v2 | 62.7 | 99.5 |95.8 |41.2 |80.0 |70.5| 75.8 |58.3 |90.0 |70.4| 172,836| 23,523|
> > > NELL-995 v1   | 96.4 |  96.4 | 98.2  |80.0  |97.3 | 88.5  |97.3  |87.4  |98.8 | 89.6 | 6,900 | 662 |
> > > NELL-995 v2 | 38.7 | 100.0  |95.6  |48.3 | 68.5  |74.2  |55.1 | 65.2  | 89.7  |74.2 | 112,752  |3,575  |
> > > WN18RR v1 | 79.0| 100.0 |98.0 |62.8 |88.7 |81.4| 87.5| 77.1| 92.3 |80.1| 2,592 |292 |
> > > WN18RR v2 | 62.6 | 100.0 | 99.6 |60.8 |81.2 |80.4 |76.9 | 75.6 | 92.7 | 80.3 |5,580 | 856 |
> > > WN18RR v3 | 54.8  | 100.0 |  94.1 |  28.3  | 75.7  | 64.1  | 69.3  | 44.1 |  82.8  | 64.0 |  10,680  | 2,423 |
> > > WN18RR v4 | 74.0 |100.0 |98.3 |58.0| 86.4| 79.0| 84.5 |73.4| 94.4 |79.0 |6,000 |409|
> > >
> > >
> > > As we can see, our approach tends to obtain higher precision scores, whereas
> > > GraIL achieves  higher recall on many of the benchmarks.
> > > The causes for low recall scores in our approach are discussed in Section 4.
> > > Overall, GraIL achieves better combined scores (F1, AUC), although
> > > our approach is still competitive. Furthermore, GraIL is much
> > > slower to train and suffers from serious scalability problems for large datasets.
> > > Finally, as stated in our previous response, our approach admits rule-based explanations
> > > for all predictions and provides precise logical guarantees, which is its
> > > main
> > > differentiating feature.
> > > Although we still feel that reporting results for DRUM and AnyBURL
> > > is sufficient to support the main claims in our paper,
> > > we would be happy to refer
> > > to the results above for GraIL in the final version.
> > >
> > > &nbsp;
> > > >MGNN transforms each pair of entities that co-occurs in a fact from the KG to a vertex in the encoded graph. However, the large number of vertices in the encoded graph may significantly increases the computational cost.
> > > The authors may want to analyze the space/time complexities of MGNN and compare with other baseline methods.
> > >
> > > The number of vertices in the encoded graph is linear in the number of triples in the KG, which
> > > is of a similar order to other embeddings in alternative approaches.
> > > In Table 2 of the paper we reported training times  and we can observe that our system
> > > is not slower to train than DRUM.
> > >
> > >
> > > &nbsp;
> > > >It is still unclear how the proposed MGNN model predict the relation between two given entities. Take Fig. 1 as an example, how to predict the relation between Crime and Punishment and The Idiot from $G_D$?
> > >
> > > Please note that this is specified in Definition 4.
> > > The predictions of an MGNN $\mathcal{M}$ on a dataset $D$ are the facts contained in the set $T_{\mathcal{M}}(D)$.
> > > Thus, to obtain the relationships between 'Crime and Punishment' and 'The Idiot' for the
> > > dataset $D$ in Figure 1,
> > > we would simply retrieve from $T_{\mathcal{M}}(D)$  all the facts of the form
> > > $R ( CrimeAndPunishment, TheIdiot)$ for $R$ a predicate.
> > >
> > > &nbsp;
> > > >The authors may want to add some descriptions of the loss function.
> > >
> > > Please note that in Section 4 of the updated draft, we specify that
> > > we use (binary, unreduced) [cross-entropy loss](https://pytorch.org/docs/stable/generated/torch.nn.BCELoss.html) with a logistic sigmoid on $\lambda^L(v)$ as the output probability.
> > > This is a standard loss function.
> > > If there are any other details that the reviewer feels would be interesting to add, we will be happy to include them.

---

### Official Review · Reviewer_AzED · 2021-11-02

**Correctness:** 4
**Technical Novelty And Significance:** 3
**Empirical Novelty And Significance:** 3
**Recommendation:** 6
**Confidence:** 4

**Main Review:**

The proposed method is very interesting since it provides a theoretically motivated model that is able to learn tasks on knowledge graphs, having the possibility to extract explanations. The MGNN just requires to apply simple constraints on the original GNN architecture to guarantee monotonicity in the implemented function (positive weights, monotonic activation functions, use of the max as aggregation operator for neighbor nodes). The most valuable property is the possibility to derive the datalog rules that the model implements, as guaranteed by the provided theoretical results.

Despite being generally a good quality contribution, in my own opinion, the paper suffers from some weaknesses.

- Some parts are quite hard to follow. Even if the main ideas can be understood and are explained with sufficient clarity, the more theoretical part is quite hard to follow (in particular, Section 3, that introduces the main theoretical results on the extraction of equivalent datalog rules from a trained MGNN).  Some concepts would need a more detailed description, with examples (figures, algorithms) to provide a mode immediate understanding. For instance, the extraction process is supported by theorem 8 but it requires the enumeration of all rules. In the experimental section it is described that some constraints are imposed on the search strategy to keep the complexity low. A schema of the generic procedure, that can be implemented for the rule extraction, would have given a more compact and clear view of the proposed method and its potential limitation with respect to complexity. Other important concepts are also difficult to grasp without simple practical examples. For instance, primal graphs and in particular (c,d)-tree-like structures for variables are important concepts to undestand the limitations of the proposed architecture (the two provided examples are for instance useful). If you look at defintion 9 and what follows (I guess that the two following paragraphs are not part of the definition and should not be in italics), many imporant concepts are introduced  that involve the rule structure and the representational power of the MGNN model. This is just in few lines... (btw, what is the dimension k? The numer of internal units in the GNN?)
- The limitations of the representational power of the MGNN model should be discussed more deeply. Theorem 10 is claimed to provide results on the expressiveness of the model, but beside the proof in the appendix and a few  lines after definition 9, this aspect is not detailed to make the reader understand the actual limitations in practice. The evaluation section reports a few rules that are extracted, but they are quite simple and limited to symmetry, inverse relations and subsumption. It would be useful to provide more examples of extracted rules.
- The extaction process requires to enumerate all rules and to check them on the trained model by applying proposition 7. In the evaluation some restrictions on the considered rules are applied (rules with at most two body atoms). The issues about complexity and the possible solutions should be clearly pointed out.
- In the evaluation section there are not enough details on the used architecture (f.i. number of layers, activation function, etc). As far as I understand, the results are obtained on only one run for each dataset (dataset split/weight initialization). It would be useful to show how the architecture affects the results and how the training is stable (however this is not a major weakness since the paper contribution may be considered theoretical).

Minor remarks.
- When describing the MGNN model (definition 3) the notation of the dimension may be confusing because of the stacking of the layers. Is {\bf b}^{\ell} of dimesion n^\ell o m^\ell?
- Does the encoding of the KG consider only the pairs of constants found in the binary predicates in KG? In case, may it happen that pairs not present in the graph are needed?
- As far as I understand, all predicates (binary/unary) are considered in the vector attached to each node being it a single constant or a pair of constants. Is this just a choice to simplify the notation in the theoretical part or is it also used in the implementation? (I did not check the code).

**Summary Of The Paper:**

The paper proposes the Monotonic Graph Neural Network (MGNN) model that can be exploited to learn tasks on knowledge graphs. Knowledge graphs are referred to as "datasets", that are based on constants, unary and binary predicates. A method is defined to encode a given dataset (i.e. a KG) as a colored graph, whose nodes correspond to the constants and pairs of constants. Edges of one out of four different "colors" (types) are added to encode the relationships between constants and the occurrences of their pairs in binary predicates. A vector is stored in each node to represent the truth table of each predicate when computed on the corresponding grounding. The MGNN processes such a graph to computed a new KG in which new facts are inferred. A theoretical result shows that an equivalent set of Datalog rules can be derived from the trained MGNN, thus providing a direct explanation of the learned model.
The proposed approach is evaluated on KG benchmarks, showing promising results when compared to other 2 state of the art methods.

**Summary Of The Review:**

The paper provides an interesting model to learn tasks on knowledge graphs. However, there are some aspects (limitations of the model and of the extraction procedure, settings of the evaluation) that would need more details and/or discussion. Finally the are some parts that are dense in concepts/theory and are hard to follow (more examples/schemes would be useful to improve readability).

---

> ### Author Response · Authors · 2021-11-15
> **Response to reviewer AzED - part 1 of 2**
>
> We thank the reviewer for their valuable comments. We next discuss in detail how we have addressed them, and answer the questions raised in their review. We will release a revised version of the paper in a few days, with all changes discussed below.
>
> &nbsp;
> >Some parts are quite hard to follow. Even if the main ideas can be understood and are explained with sufficient clarity, the more theoretical part is quite hard to follow (in particular, Section 3, that introduces the main theoretical results on the extraction of equivalent datalog rules from a trained MGNN). Some concepts would need a more detailed description, with examples (figures, algorithms) to provide a mode immediate understanding.
>
> We have added more examples in Sections 2 and 3 illustrating the theoretical concepts that we introduce (please see details below).
>
> &nbsp;
> >For instance, the extraction process is supported by theorem 8 but it requires the enumeration of all rules. In the experimental section it is described that some constraints are imposed on the search strategy to keep the complexity low. A schema of the generic procedure, that can be implemented for the rule extraction, would have given a more compact and clear view of the proposed method and its potential limitation with respect to complexity.
>
> The rule extraction procedure sketched after Theorem 8,
> where all rules  of a certain syntactic shape are enumerated and individually checked,
> shows that extracting the program $\mathcal{P}_{\mathcal{M}}$
> from an MGNN $\mathcal{M}$ is algorithmically possible. This procedure
> is not meant
> to be implemented as described, and  can be optimised in a number of
> ways.
>
> For example,
>  the relevant space of rules can be significantly reduced during the execution of the algorithm
>  by exploiting the fact that rules can be subsumed (i.e., logically implied) by other rules;
>  for instance, if we have determined that
>  rule $A(x) \rightarrow  B(x)$ has been captured then
>  rules of the form  $A(x) \wedge C \rightarrow B(x)$, with $C$ any conjunction of atoms, do not need to be considered
> since such rules are logically implied by the previous one and are hence redundant.
>
> We clarify this in the revised version of the paper, and we include a pseudocode of the optimised algorithm in the Appendix.
> Note also that our implementation uses the optimised version.
>
>
> &nbsp;
> > Other important concepts are also difficult to grasp without simple practical examples. For instance, primal graphs and in particular (c,d)-tree-like structures for variables are important concepts to understand the limitations of the proposed architecture (the two provided examples are for instance useful). If you look at definition 9 and what follows (I guess that the two following paragraphs are not part of the definition and should not be in italics), many important concepts are introduced that involve the rule structure and the representational power of the MGNN model. This is just in few lines...
>
>
> The notion of primal graph is a standard concept in graph theory (e.g., see textbook [1, Chapter 9]), which is regularly applied to data management theory (e.g., see [2, Chapter 6]).
> In the revised version of the paper, we expand the examples presented after Definition 9 and show explicitly the primal graph of each rule given as an example.
>
> [1] Rina Dechter: Constraint Processing. Elsevier Morgan Kaufmann, 2003.
>
> [2] Serge Abiteboul, Richard Hull, Victor Vianu: Foundations of Databases. Addison-Wesley, 1995.
>
> &nbsp;
> >I guess that the two following paragraphs are not part of the definition and should not be in italics.
>
>
> Please note that Definition 9 has three paragraphs, which should all be in italics. In particular, the second paragraph defines the notion of a $(c,d)$-tree-like rule, and the third one defines $\mathcal{P}_{\mathcal{M}}$.
>
> &nbsp;
> >What is the dimension k? The numer of internal units in the GNN?
>
>
> The dimension $k$ of an MGNN is defined in the last sentence of Definition 3: it is the maximum column dimension for a matrix in the MGNN. In the revised version of the paper, we add a reminder about this definition before Definition 9.

---

> ### Author Response · Authors · 2021-11-15
> **Response to reviewer AzED - part 2 of 2**
>
> _(this continues our previous comment, which contains part 1 of the response)_
>
> &nbsp;
> >The limitations of the representational power of the MGNN model should be discussed more deeply. Theorem 10 is claimed to provide results on the expressiveness of the model, but beside the proof in the appendix and a few lines after definition 9, this aspect is not detailed to make the reader understand the actual limitations in practice. The evaluation section reports a few rules that are extracted, but they are quite simple and limited to symmetry, inverse relations and subsumption. It would be useful to provide more examples of extracted rules.
>
>
> In the revised version of the paper we discuss in more detail the kinds of rules
> that MGNNs can capture (see the paragraphs following Definition 9). Furthermore, we give a more
> detailed discussion of each example of a rule that can be captured by an MGNN.
>
> For more examples, please note that the supplemental material includes a list of all rules captured in benchmark WN18RRv1 in _rules/extracted/GraIL-BM\_WN18RR\_v1\_from-data\_EC\_0\_extrarules.txt_, and it also includes detailed instructions
> in _README.md_ for extracting rules from trained models for all other benchmarks.
>
> &nbsp;
> >The extaction process requires to enumerate all rules and to check them on the trained model by applying proposition 7. In the evaluation some restrictions on the considered rules are applied (rules with at most two body atoms). The issues about complexity and the possible solutions should be clearly pointed out.
>
>
> Please refer to our previous comment, where we clarify that the extraction procedure described after Theorem 8 is  implemented not directly, but with optimisations. In the Appendix of the revised version of the paper we give the pseudocode of the optimised algorithm, discuss the complexity of this procedure, and show that its output is a program $\mathcal{P}$  which corresponds to  $\mathcal{P}_{M}$ (from Definition 9) without redundant rules.
>
> &nbsp;
> >In the evaluation section there are not enough details on the used architecture (f.i. number of layers, activation function, etc). As far as I understand, the results are obtained on only one run for each dataset (dataset split/weight initialization). It would be useful to show how the architecture affects the results and how the training is stable (however this is not a major weakness since the paper contribution may be considered theoretical).
>
>
> We add these details in the revised version of the paper.
>
> &nbsp;
> >When describing the MGNN model (definition 3) the notation of the dimension may be confusing because of the stacking of the layers. Is ${\bf b}^{\ell}$ of dimension $n^\ell$ o $m^\ell$?
>
> Definition 3 uses consistent notation for dimensions, with letter $m$ always used for row dimension, and letter $n$ always used for column dimension. Please note that Definition 3 states that $\mathbf{b}^{\ell}$ is a vector of dimension $n^{\ell}$.
>
> &nbsp;
> >Does the encoding of the KG consider only the pairs of constants found in the binary predicates in KG?
>
>
> Yes. Definition 2 implies that if there is a vertex $v_{a,b}$ in an encoding of dataset, there must be a fact of the form $S(a,b)$ or $S(b,a)$ in the dataset. This ensures efficiency of the encoding.
>
> &nbsp;
> > In this case, may it happen that pairs not present in the graph are needed?
>
>
> We discuss this possibility in Section 4, in the *Evaluation results* subsection. This possibility is precisely what motivates the experiment summarised in Table 3.
>
> &nbsp;
> >As far as I understand, all predicates (binary/unary) are considered in the vector attached to each node being it a single constant or a pair of constants. Is this just a choice to simplify the notation in the theoretical part or is it also used in the implementation? (I did not check the code).
>
> The encoding used in the implementation is the encoding presented in the paper.
> Our encoding attaches vectors of equal dimension to all nodes to simplify the presentation.
> However, please note that all elements of feature vectors in the intermediate layers of the MGNN (which may be different from $\delta$) can be used to represent latent features of a node, regardless of whether the node represents a single constant or a pair.

---

### Author Response · Authors · 2021-11-18
**Revised version of the paper**

We have uploaded a revised version of the paper with all modifications discussed in the reviews.

The most significant changes are:
* adding new examples in Sections 2 and 3, in particular a figure showing the encoding (and decoding) of a dataset into a coloured graph;
* adding a new appendix describing the rule extraction algorithm that we use in Section 4;
* adding all requested details about the experimental evaluation in Section 4; and
* adding all suggested references to the Related Work section.

---

> ### Author Response · Authors · 2021-11-22
> **New version of the paper.**
>
> We have uploaded a new version of the paper in response to the updated reviews. The two main changes are:
> * We reference the explanatory capabilities of modern Datalog engines, in Section 1.
> * We include a new experiment in Section 4, where we evaluate the programs extracted from MGNN models.

---

### Decision · Program_Chairs · 2022-01-20

**Decision:**

Accept (Poster)

**Comment:**

This paper proposes monotonic graph neural networks (MGNNs) for the transformation of knowledge graphs. Specifically, MGNNs transform a knowledge graph into a colored graph where each node is represented by a numeric feature vector and each edge encodes the node relationship with different colors. The authors provide theoretical analysis showing that monotonic constraint can enable the model to derive logical inference rules in Datalog, and thus the trained model is explainable.


The authors addressed most of the concerns raised by the reviewers, such as motivation, runtime, and comparison with existing baselines. Three of the four reviewers are positive (with the scores of 6 or above) towards acceptance after rebuttal discussions, and the remaining reviewer gives a score of 5 (below acceptance threshold) thinks that this work still lacks novelty, but he/she is not against acceptance if other reviewers choose so. Considering this work makes a good exploration on explainable graph neural networks, which is an interesting and important research direction, we recommend for acceptance. We thank the reviewers and the authors for their active discussion.